# Reward and Guidance through Rubrics: Promoting Exploration to Improve Multi-Domain Reasoning

**Baolong Bi**[1][2]   **Shenghua Liu**[†][1][2]   **Yiwei Wang**[3]   **Siqian Tong**[2]   **Lingrui Mei**[1][2]
**Yuyao Ge**[1][2]   **Yilong Xu**[2]   **Jiafeng Guo**[1][2]   **Xueqi Cheng**[1][2]

## Abstract

Recent advances in reinforcement learning (RL) have significantly improved the complex reasoning capabilities of large language models (LLMs). Despite these successes, existing methods mainly focus on single-domain RL (e.g., mathematics) with verifiable rewards (RLVR), and their reliance on purely online RL frameworks restricts the exploration space, thereby limiting reasoning performance. In this paper, we address these limitations by leveraging rubrics to provide both fine-grained reward signals and offline guidance. We propose **RGR-GRPO** (**R**eward and **G**uidance through **R**ubrics), a rubric-driven RL framework for multi-domain reasoning. RGR-GRPO enables LLMs to receive dense and informative rewards while exploring a larger solution space during GRPO training. Extensive experiments across 14 benchmarks spanning multiple domains demonstrate that RGR-GRPO consistently outperforms RL methods that rely solely on alternative reward schemes or offline guidance. Compared with verifiable online RL baseline, RGR-GRPO achieves average improvements of +7.0%, +5.4%, +8.4%, and +6.6% on mathematics, physics, chemistry, and general reasoning tasks, respectively. Notably, RGR-GRPO maintains stable entropy fluctuations during off-policy training and achieves superior pass@k performance, reflecting sustained exploration and effective breakthrough beyond existing performance bottlenecks.

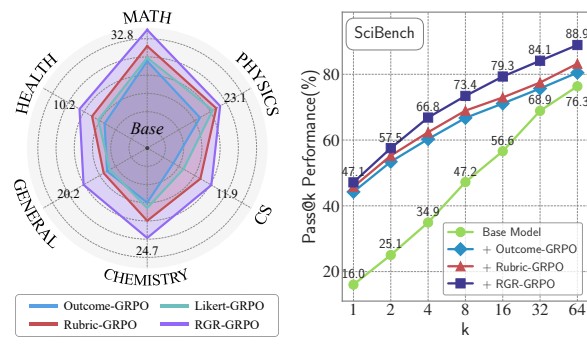

(a) Improvements (%) across Multiple Domains     (b) Pass@k Performance on SciBench

*Figure 1.* Our RGR-GRPO shows strong cross-domain reasoning capability and expands the frontier of exploration.

## 1. Introduction

Reinforcement learning (RL) has emerged as a core post-training paradigm that has substantially advanced the reasoning capabilities of large language models (LLMs) (Yang et al., 2025a; Jaech et al., 2024; Team et al., 2025), spanning tasks such as scientific reasoning (Burgess et al., 2025), medical question answering (Arora et al., 2025), and code generation (Yang et al., 2025b). Among recent breakthroughs, Reinforcement Learning with Verifiable Rewards (RLVR) has been particularly effective: by leveraging rule-based and automatically verifiable rewards, RLVR enables LLMs to acquire complex reasoning skills through trial-and-error exploration. Notably, R1-Zero (Guo et al., 2025) demonstrates that directly training a base LLM with explicit scalar rewards (e.g., correctness and format) can yield impressive reasoning capabilities without supervised fine-tuning.

Despite these successes, current RL approaches (Zeng et al., 2025; Yu et al., 2025) face two key limitations: **(1) Domain-limited and sparse rewards.** Most methods rely on verifiable single-domain tasks such as mathematics or coding, where rule-based checking provides precise but sparse supervision. However, open-ended multi-domain reasoning tasks often lack standard answers, making reward design difficult to generalize and reducing training efficiency (Cui et al., 2025). **(2) Restricted online exploration.** Purely online RL frameworks explore within a narrow policy space, constrained by limited on-policy samples and short-horizon updates. This restricted exploration prevents models from

---

[1] State Key Laboratory of AI Safety, Institute of Computing Technology, CAS [2]University of Chinese Academy of Sciences [3]University of California, Merced. Correspondence to: Baolong Bi, Shenghua Liu <{bibaolong23z,liushenghua}@ict.ac.cn>.

*Proceedings of the $43^{rd}$ International Conference on Machine Learning*, Seoul, South Korea. PMLR 306, 2026. Copyright 2026 by the author(s).

effectively leveraging diverse reasoning trajectories or discovering higher-quality solutions beyond the immediate reward signals (Zhao et al., 2025; Yue et al., 2025).

In this paper, we propose to address these challenges by introducing rubrics into the GRPO (Shao et al., 2024) training process to provide reliable dense rewards and guide offline rollout refinement. Through preliminary experiments, we show that rubrics can provide dense and informative rewards, leading to fewer non-advantageous trajectories, and that rubric-guided self-refinement at test time enables continuous improvement across different training stages. Building on these observations, we introduce **RGR-GRPO** (**R**eward and **G**uidance through **R**ubrics), a rubric-driven reinforcement learning framework for multi-domain reasoning. RGR-GRPO enhances reasoning capability by supporting dense rubric-based rewards and off-policy guidance on suboptimal trajectories, thereby promoting more effective exploration across diverse domains. Specifically, RGR-GRPO consists of two key components:

- **Rubric-based fine-grained rewards.** To overcome the limitations of single-domain training and sparse reward signals, RGR-GRPO constructs question-specific rubrics that span multiple domains. Each rubric comprises two complementary types of evaluation criteria: **Factual** and **Process**, each assigned an adaptive weight reflecting its relative importance. The Factual criteria verify the accuracy of intermediate and final results, whereas the Process criteria measure the logical soundness of the reasoning trajectory. Under an LLM-as-Judge setup, RGR-GRPO evaluates generated rollouts against these criteria and aggregates the weighted scores into scalar rewards. This rubric-based reward design provides reliable and fine-grained feedback, enabling RLVR to generalize across diverse domains beyond mathematics and code.

- **Rubric-guided offline exploration.** To break the exploration bottleneck of purely online methods, RGR-GRPO further integrates rubric-based signals into an off-policy guidance mechanism, following recent advances in off-policy approaches (Yan et al., 2025; Zhang et al., 2025b). Specifically, RGR-GRPO identifies high-reward rollouts under the current policy and analyzes their imperfect criteria based on rubric evaluation. These partial deficiencies are then used as targeted offline guidance, prompting the policy model to self-refine and generate improved reasoning trajectories. Through this process, rubrics effectively expand the exploration space of on-policy RL process, bridging the gap between dense reward learning and structured offline refinement.

To compare RGR-GRPO with verifiable sparse-reward methods, we conduct Zero-RL training using the **WebInstruct-verified** (Ma et al., 2025) dataset, a high-quality and diverse-domain benchmark for verifiable reasoning tasks.

We train Qwen2.5-3B and 7B models (Yang et al., 2025a) and evaluate them on a wide range of benchmarks spanning multiple scientific domains (mathematics, physics, and chemistry) as well as general reasoning benchmarks. During training, purely online RL methods exhibit early entropy collapse, causing the policy to converge to limited trajectories. In contrast, RGR-GRPO, guided by offline rubric-based supervision, maintains a smoother and more gradual entropy decline—indicating sustained exploration and continual learning. Experimental results show that RGR-GRPO consistently outperforms all baselines. On the 7B model, RGR-GRPO achieves average improvements of 7.0%, 5.4%, and 8.4% in mathematics, physics, and chemistry domains, respectively, and 6.6% in general-domain reasoning, compared with outcome-based reward methods. Moreover, RGR-GRPO exhibits more stable training dynamics than other off-policy baselines, maintaining sustained exploration without premature collapse or entropy explosion. Notably, across different pass@k settings on SciBench, RGR-GRPO continues to outperform standard RL, demonstrating its ability to break the exploration bottleneck and significantly enhance multi-domain reasoning capabilities.

## 2. RGR-GRPO: Reward and Guidance through Rubrics

### 2.1. Preliminary

Group Relative Policy Optimization (GRPO) (Shao et al., 2024) is an online RL algorithm that extends the Proximal Policy Optimization (PPO) (Schulman et al., 2017) framework while removing its dependency on a separate value function. Instead of estimating token-level advantages through a critic, GRPO evaluates the relative performance among a group of sampled responses for the same query. Concretely, given a question $q$ drawn from the training distribution $\mathcal{D}$, the old policy $\pi_{\theta_{\text{old}}}$ generates a group of $G$ responses $\{o_i\}_{i=1}^{G}$. Each response is assigned a reward score $r_i$ by the reward model, and these scores are normalized within the group to compute the relative advantage:

$$\hat{A}_i = \frac{r_i - \text{mean}\left(\{r_j\}_{j=1}^{G}\right)}{\text{std}\left(\{r_j\}_{j=1}^{G}\right)}, \quad (1)$$

GRPO then updates the policy parameters $\theta$ by:

$$\mathcal{J}_{\text{GRPO}}(\theta) = \mathbb{E}_{q \sim \mathcal{D}, \{o_i\}_{i=1}^{G} \sim \pi_{\theta_{\text{old}}}(\cdot|q)} \frac{1}{G} \sum_{i=1}^{G} \frac{1}{|o_i|} \sum_{t=1}^{|o_i|} \left\{ \min\left[ \right.\right.$$

$$\left.\left. r_t^{(i)}(\theta)\hat{A}_i, \text{CLIP}\left(r_t^{(i)}(\theta), 1-\epsilon, 1+\epsilon\right)\hat{A}_i \right] - \beta D_{\text{KL}}(\pi_\theta|\pi_{\text{ref}}) \right\},$$

$$(2)$$

where $r_t^{(i)}(\theta) = \frac{\pi_\theta(o_i|q, o_{<t}^{(i)})}{\pi_{\theta_{\text{old}}}(o_i|q, o_{<t}^{(i)})}$ serves as an importance sampling ratio that corrects the gradient estimation according to policy gradient theory (Sutton et al., 1999).

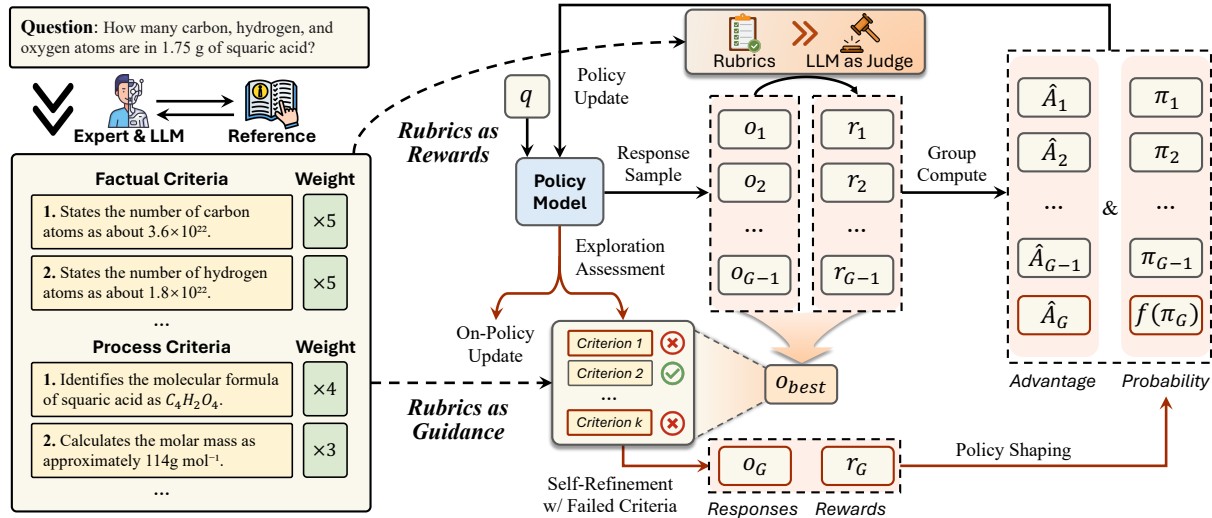

*Figure 2.* Overview of the RGR-GRPO framework: (a) Construct rubrics for RL reward based on the input question and reference answer. (b) Conduct exploration assessment with the best response $o_{best}$, determining whether off-policy guidance is required. When exploration is insufficient, failed criteria are then used to refine $o_{best}$ into off-policy rollouts, and the sampling probabilities are reshaped via a shaping function to update the policy model.

$\epsilon$ and $\beta$ are hyperparameters controlling the clipping range and the KL divergence regularization, respectively. These constraints help maintain the updated policy $\pi_\theta$ within a stable region around the previous policy $\pi_{\theta_{old}}$, effectively preventing abrupt policy shifts. This formulation characterizes an on-policy reinforcement learning setup, where optimization relies on samples generated from a distribution closely aligned with the current policy.

### 2.2. Rubric-based Fine-Grained Rewards

While rubric-based rewards have been adopted in certain RL frameworks (Viswanathan et al., 2025; Gunjal et al., 2025; Jayalath et al., 2025), unconstrained rubric design often risks reward hacking or misalignment with true reasoning task objectives (Skalse et al., 2022; Gao et al., 2023). To address this, we design two complementary types of rubrics tailored for complex, multi-domain reasoning tasks:

- **Factual Criteria:** Verify the correctness of intermediate reasoning, sub-answers, and final results.
- **Process Criteria:** Evaluate whether the reasoning process follows essential steps and valid logic.

We generate the rubrics in a two-step process using an expert LLM, *OpenAI O3* (Jaech et al., 2024). First, for each question $q$, we prompt *O3* to generate a high-quality reference answer $a_{ref}$ that contains a comprehensive solution, including detailed reasoning traces and intermediate verification outcomes. We then filter out any cases where the reference solution's final answer is judged incorrect, ensuring the reliability of the retained references. Second, we instruct *O3* to generate the fine-grained rubrics $\mathcal{C} = \{c_k\}_{k=1}^{|\mathcal{C}|}$

by conditioning on both the original question $q$ and the generated $a_{ref}$. This grounding ensures the rubrics accurately reflect the correct reasoning process and sub-answers required for the question. Each criterion $c_k$ is composed of a descriptive specification $d_k$ and and an adaptive weight $w_k$, i.e., $c_k = (d_k, w_k)$. A simple example is illustrated in Figure 2 (a), with prompt for rubric generation in Appendix E.3.

For each criterion $c_k \in \mathcal{C}$ associate with question $q$, we evaluate the model output $o_i$ using a judge model that produces a binary verification score:

$$s_k(q,o_i) = \begin{cases} 1, & \text{if response } o_i \text{ satisfies } d_k \text{ given prompt } q, \\ 0, & \text{otherwise.} \end{cases}$$

(3)

The binary score $s_k(q,o_i) \in \{0,1\}$ effectively mitigates the risk of reward hacking while maintaining interpretability. We then aggregate all rubric scores along with their corresponding weights $\{(w_k, s_k)\}_{k=1}^{|\mathcal{C}|}$ into a normalized scalar reward:

$$R(q,o_i) = \frac{\sum_{k=1}^{n} w_k \cdot s_k(q,o_i)}{\sum_{k=1}^{n} w_k},$$

(4)

To allow the policy to produce correct answers even when its reasoning process deviates from the predefined rubrics, we consider the reward to be fully satisfied if all factual criteria $\mathcal{C}_i^{fact}$, including both final and intermediate sub-answers, are verified. Process rewards are considered only when factual verification fails, allowing the model to explore reasoning trajectories beyond a single reference path. The final reward

for a given query-answer pair $(q,o_i)$ is then computed as:

$$r_i = \begin{cases} 1, & \sum_{k=1}^{|\mathcal{C}_i^{\text{fact}}|} s_k(q,o_i) = |\mathcal{C}_i^{\text{fact}}| \text{ where } c_k \in \mathcal{C}_i^{\text{fact}}, \\ R(q,o_i), & \text{otherwise.} \end{cases}$$

(5)

Separating factual correctness and essential reasoning steps into distinct rubrics simplifies the evaluation process for the judge model, thereby enhancing the reliability of LLM-as-a-judge assessments. The rubric-based binary verification further provides both reliable and dense reward signals, striking a balance between mitigating reward hacking and reducing intra-group variance. Incorporating this reward into Eq. (1 - 2) for online RL, as shown in Section 5.1, leads to consistent performance gains over alternative reward formulations.

## 2.3. Rubric-Guided Offline Exploration

Recent studies (Yan et al., 2025; Zhang et al., 2025b) have demonstrated the potential of off-policy guidance in enhancing reinforcement learning. Building on the effectiveness of Rubric-Guided Self-Refinement in improving model reasoning during test time (Section 5.2), we further incorporate rubrics as an off-policy guidance signal into the original GRPO framework to expand the model's exploration boundary. Specifically, our RGR-GRPO framework consists of three steps, as illustrated in Figure 2 (b):

**Step 1: Exploration Assessment.** In each GRPO training iteration, we first sample $G-1$ initial responses $\{o_i\}_{i=1}^{G-1}$ for each question $q$ from the old policy $\pi_{\theta_{\text{old}}}$, while reserving the last rollout for adaptive exploration adjustment. Each response is evaluated by the rubric-based reward function (Section 2.2), producing both the aggregated reward and detailed criterion judgments:

$$r_i, \{s_k(q,o_i)\}_{k=1}^{|\mathcal{C}_i|} \leftarrow \text{Reward}(q,o_i), \quad \forall i \in [1, G-1]. \quad (6)$$

The goal of off-policy guidance is to overcome the exploration limitations of purely on-policy updates. If the current group of responses already contains a perfect solution, additional off-policy exploration is unnecessary, which helps prevent excessive distributional shift and subsequent training collapse (Zhang et al., 2025b). Specifically, we locate the best-performing response $o_{\text{best}}$ as:

$$o_{\text{best}} = \underset{o_i \in \{o_1, \ldots, o_{G-1}\}}{\arg\max} r_i, \quad (7)$$

We then determine the update strategy according to whether $o_{\text{best}}$ meets all rubric criteria:

- **If** $\sum_{k=1}^{|\mathcal{C}_i|} s_k(q, o_i) = |\mathcal{C}_i|$: the on-policy exploration suffices, so we generate the final response $o_G$ and update the policy via Eq. (1 - 2).
- **Otherwise:** the policy fails to reach a perfect solution, and subsequent mix-policy refinement is applied.

Exploration Assessment (EA) can determine whether off-policy guidance is needed based on the current group's exploration upper bound, effectively avoiding unnecessary off-policy updates and reducing the risk of entropy explosion. A more detailed proof and analysis are provided in Appendix C.

**Step 2: Rubric-Based Self-Refinement.** To enhance the upper bound of the current exploration group, we refine the best response $o_{\text{best}}$ by explicitly conditioning on its failed rubric items. Inspired by Critique-GRPO (Zhang et al., 2025b), we prompt the policy model with the triplet $(q, o_{\text{best}}, \mathcal{C}^{\text{failed}})$, where

$$\mathcal{C}^{\text{failed}} = \{c_k \mid s_k(q, o_{\text{best}}) = 0\} \quad (8)$$

denotes the set of unsatisfied criteria. Each failed criterion $c_k \in \mathcal{C}^{\text{failed}}$ is concatenated in order, and a self-refining template $T_{\text{refine}}$ (Appendix E.2) is used to generate a refined response:

$$o_G \sim \pi_{\theta_{\text{old}}}(\cdot \mid q, o_{\text{best}}, \mathcal{C}^{\text{failed}}), \quad (9)$$

and its corresponding reward $r_G$ is computed accordingly.

**Step 3: Mix-Policy GRPO.** Finally, we merge the off-policy refined rollout with the initial on-policy rollouts to jointly update the policy. The advantage estimation still follows Eq. (1), covering all responses in the batch. The model is then optimized under a mixed-policy objective adapted from GRPO:

$$\mathcal{J}_{\text{RGR-GRPO}}(\theta) = \mathbb{E}_{q \sim \mathcal{D}, \{o_i\}_{i=1}^{G-1} \sim \pi_{\text{old}}, o_G \sim \pi_{\text{old}}(\cdot \mid q, o_{\text{best}}, \mathcal{C}^{\text{failed}})}$$

$$\frac{1}{G} \left[ \underbrace{\sum_{i=1}^{G-1} \sum_{t=1}^{|o_i|} r_t^{(i)}(\theta) \hat{A}_i}_{\text{on-policy objective}} + \underbrace{\sum_{t=1}^{|o_G|} f_{shape}\left(r_t^{(G)}(\theta)\right) \hat{A}_G}_{\text{off-policy objective}} \right],$$

(10)

Here, $r_t^{(i)}(\theta) = \frac{\pi_\theta(o_i|q,o_{<t}^{(i)})}{\pi_{\theta_{\text{old}}}(o_i|q,o_{<t}^{(i)})}$ is the normal importance sampling ratio as defined in Eq. (2). The off-policy refinement term is further modulated by a shaping function $f(\cdot)$ (Yan et al., 2025) to adjust the contribution of each token in off-policy refined response $o_G$:

$$f_{shape}\left(r_t^{(G)}(\theta)\right) = \frac{\pi_\theta(o_G \mid q, o_{<t}^{(G)})}{\pi_\theta(o_G \mid q, o_{<t}^{(G)}) + \gamma}. \quad (11)$$

where $0 < \gamma < 1$. The shaping function reweights the gradients by assigning higher importance to low-probability tokens in the refined trajectories, encouraging the model to learn from successful but out-of-distribution behaviors while mitigating the impact of failed refinements.

Overall, RGR-GRPO effectively balances on-policy stability with off-policy exploration flexibility, leading to more robust and generalizable policy optimization.

| Model | Math | | | Physics | | Chemistry | | General | | | | | AVG. |
|---|---|---|---|---|---|---|---|---|---|---|---|---|---|
| | MATH | MATH500 | SMath | PIQA | SPhys | Chem | SChem | MMLU | MMLU$^+$ | GPQA$^*$ | GPQA | OLY | |
| *Qwen2.5-3B* | | | | | | | | | | | | | |
| Base-Model | 59.0 | 49.2 | 32.0 | 79.9 | 18.9 | 36.1 | 17.3 | 57.1 | 30.0 | 24.3 | 26.7 | 10.9 | 36.8 |
| Instruct-Model | 63.8 | 53.6 | 49.0 | **83.0** | 22.5 | 38.5 | 26.3 | **66.3** | **42.3** | 28.3 | 30.4 | **23.7** | 43.9 |
| Outcome-GRPO$^\dagger$ | 64.3 | 55.7 | 44.9 | 79.7 | 26.0 | **40.9** | 30.1 | 61.3 | 39.8 | 27.3 | 25.2 | 20.1 | 42.9 |
| Likert-GRPO$^\dagger$ | 64.5 | 54.8 | 49.0 | 78.5 | 28.8 | 39.2 | 28.9 | 62.6 | 37.8 | 29.8 | 28.6 | 18.7 | 43.4 |
| Rubric-GRPO$^\dagger$ | 63.4 | 56.0 | 47.6 | 80.6 | 27.9 | 39.7 | 28.2 | 63.2 | 38.4 | 27.8 | 29.6 | 20.3 | 43.6 |
| Critique-GRPO$^\ddagger$ | 61.5 | 52.4 | 43.5 | 79.3 | 24.2 | 38.2 | 26.2 | 57.5 | 33.7 | 25.3 | **31.9** | 17.3 | 40.9 |
| LUFFY$^\ddagger$ | 62.3 | 54.4 | 41.5 | 79.5 | 24.7 | 38.8 | 25.6 | 60.3 | 34.3 | 27.8 | 29.5 | 18.4 | 41.4 |
| RGR-GRPO$^\ddagger$ (Ours) | **66.3** | **57.0** | **50.8** | 80.2 | **31.8** | 40.3 | **31.6** | 64.5 | 39.8 | **30.8** | 30.9 | 21.5 | **45.5** |
| *Qwen2.5-7B* | | | | | | | | | | | | | |
| Base-Model | 64.0 | 52.6 | 41.5 | 84.1 | 22.9 | 44.5 | 29.7 | 69.7 | 45.0 | 29.3 | 31.0 | 20.9 | 44.6 |
| Instruct-Model | 72.8 | 65.0 | 58.5 | **86.6** | 33.9 | 46.6 | 40.2 | 73.9 | 55.9 | 32.3 | 33.5 | 28.4 | 52.3 |
| Outcome-GRPO$^\dagger$ | 73.2 | 61.8 | 61.2 | **86.6** | 38.3 | 47.0 | 38.3 | 73.3 | 52.2 | 35.4 | 32.7 | 27.3 | 52.3 |
| Likert-GRPO$^\dagger$ | 73.5 | 62.7 | 61.5 | 84.7 | 44.5 | 47.3 | 39.1 | 71.5 | 52.0 | 34.8 | 32.6 | 28.4 | 52.7 |
| Rubric-GRPO$^\dagger$ | 73.9 | 63.8 | 65.3 | 84.6 | **45.8** | 47.9 | 40.2 | 71.0 | 53.9 | 36.4 | 35.5 | 26.2 | 53.7 |
| Critique-GRPO$^\ddagger$ | 71.4 | 64.8 | 61.2 | 84.8 | 36.6 | 45.7 | 38.7 | 71.5 | 49.1 | 27.8 | 29.9 | 25.6 | 50.6 |
| LUFFY$^\ddagger$ | 72.2 | 61.6 | 60.5 | 85.2 | 35.2 | 46.4 | 38.7 | 70.4 | 49.8 | 29.8 | 31.5 | 24.8 | 50.5 |
| RGR-GRPO$^\ddagger$ (Ours) | **75.2** | **66.8** | **67.9** | 86.3 | 45.4 | **48.8** | **43.7** | **74.3** | **56.7** | **38.9** | **36.7** | **28.9** | **55.8** |

*Table 1.* Performance (%) on subject-specific and general-domain benchmarks. On-policy and hybrid off-policy RL methods are denoted by $\dagger$ and $\ddagger$, respectively. The best result for each metric is in **bold**, and the second best is underlined.

## 3. Experimental Setup

**Datasets Construction.** To compare the effectiveness of multi-domain reasoning rewards, we construct our training set based on *WebInstruct-Verify* (Ma et al., 2025), which is a large-scale, multi-domain, and verifiable dataset covering diverse subjects such as physics, chemistry, social sciences, and finance. We randomly sample a subset and filter prompts by removing those with excessively long reference answers or overly simple examples. The final dataset contains approximately 10k samples used for both RL training and validation. Detailed statistics are provided in Appendix A.1.

**Evaluation.** We evaluate model performance across two categories: specific-subject and general reasoning tasks. The specific-subject evaluation focuses on three core scientific disciplines—mathematics, physics, and chemistry—which best reflect the model's reasoning ability within independent scientific domains. Each subject includes multiple benchmark datasets. The general reasoning category covers broader benchmarks, including MMLU (Hendrycks et al., 2020), MMLU-Pro (MMLU$^+$) (Wang et al., 2024), GPQA, GPQA-Diamond (GPQA$^*$) (Rein et al., 2023), and OlympicArena-Valid (OLY) (Huang et al., 2024). We follow Fan et al. (2025) and adopt the *Open-Science-Evaluation* framework for all tasks evaluation. During evaluation, we use greedy-decoding (temperature=0) and shuffle

multiple-choice options to avoid contamination. Further details are provided in Appendix A.3.

**Baselines.** We compare three reward mechanisms under the on-policy GRPO framework: (1) *Outcome-GRPO*, where the reward is computed through binary verification based on the final answer; (2) *Likert-GRPO*, which provides a dense reward by comparing the model output against the reference answer; and (3) *Rubric-GRPO*, where the reward is computed by aggregating the verification results from question-specific rubrics (Section 2.2). For *Outcome-GRPO*, we use *Math-Verify*[1] to extract and compute Outcome rewards. For both the *Likert-* and *Rubric-GRPO* settings, we employ *Qwen3-32B* as the judge model. For off-policy strategies, we include two baselines: LUFFY (Yan et al., 2025), which directly mixes offline supervised responses (from *OpenAI-O3*) with on-policy rollouts; and Critique-GRPO (Zhang et al., 2025b), which leverages ground-truth-based critiques to guide policy refinement, shows more stable and stronger performance than the variant using critiques from a strong model under our setting.

**RL Implementation.** All baselines and our proposed RGR-GRPO are implemented using the *Verl* framework (Sheng et al., 2025). Training is conducted for 400 steps with a batch size of 96 and a learning rate of $1 \times 10^{-6}$.

---

[1]https://github.com/huggingface/Math-Verify

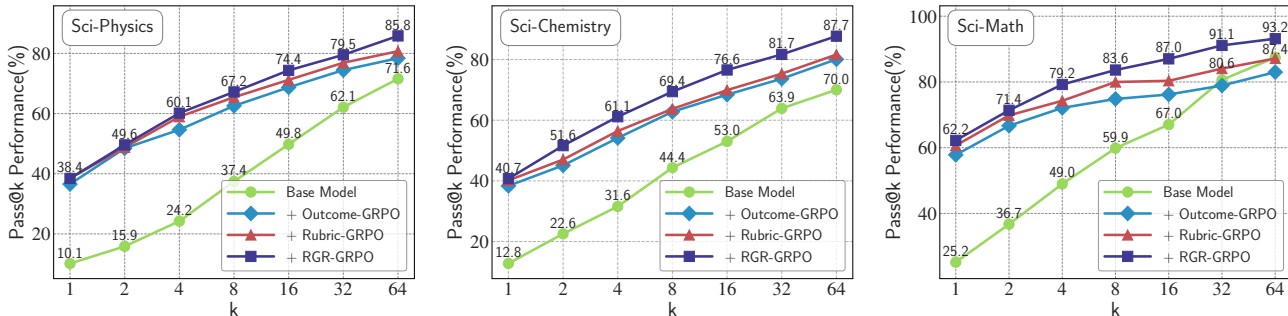

*Figure 3.* Pass@*k* performance (%) of Qwen2.5-7B across physics, chemistry, and math subjects in Sci-Bench.

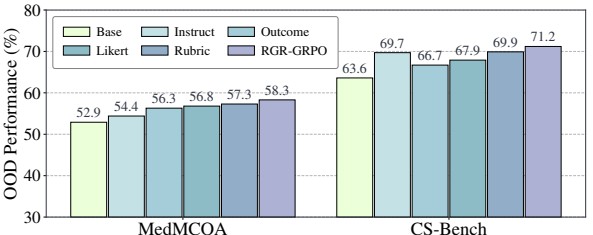

*Figure 4.* Comparison of out-of-distribution (OOD) performance (%) on the MedMCQA and CS-Bench datasets.

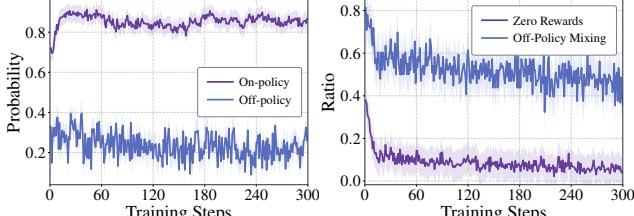

*Figure 5.* Distribution and average of file / function match rate and resolved rate on SWE-Bench Lite LeaderBoard.

We use temperature = 1.0 for rollout generation and sample 8 rollouts per prompt. We also introduce a length penalty (Liu et al., 2025b) to discourage overly long reasoning: $r_i = r_i - \lambda(L_i - L^*)$, with $\lambda = 1 \times 10^{-4}$ and $L^* = 2k$. For on-policy baselines, we follow *Verl*'s default GRPO configuration with 8 on-policy rollouts. For RGR-GRPO and all off-policy baselines, we use 1 off-policy and 7 on-policy rollouts to ensure fair comparison. Following prior work, we set the shaping coefficient $\gamma = 0.1$ and apply an entropy coefficient of 0.01 to encourage exploration. To enable more flexible policy updates, we remove both the clipping function for probability ratios and the KL-divergence constraint by setting $\beta = 0$ from the original GRPO formulation, thereby encouraging more substantial model adaptation and more effective learning from refinement signals. In addition, recent studies (Liu et al., 2025b) suggest that token-level normalization and standard-deviation scaling in advantage estimation can introduce biased optimization; we thus omit these components to ensure a more stable and unbiased objective.

## 4. Experimental Results

### 4.1. Main Results

We conduct 400-step RL training on both the *Qwen2.5-3B* and *Qwen2.5-7B* base models, saving checkpoints every 40 steps for evaluation. For each method, we report the best checkpoint performance in Table 1. Under the pure on-policy setting, the Rubric-GRPO—whose rubrics consist of Factual and Process criteria—achieves the highest average performance, demonstrating that our rubric design provides reliable and verifiable dense rewards for effective RL optimization. In contrast, the off-policy baselines Critique-

GRPO and LUFFY fail to achieve consistent improvements, yielding smaller gains than the on-policy methods.

We further observe that both methods exhibit instability during training, with entropy explosions, which we attribute to excessive distributional shifts caused by divergent off-policy data (see Appendix D for analysis). In comparison, our RGR-GRPO maintains stable hybrid-policy training guided by explicit rubrics and consistently delivers the best results. On the 7B model, RGR-GRPO outperforms the base model by an average of 25.1%, the official instruct model by 6.7%, and the second-best on-policy Rubric-GRPO by 3.7%. These results demonstrate that incorporating rubric-guided off-policy refinements significantly improves exploration efficiency and leads to the best overall performance.

### 4.2. Out-of-distribution Performance

We further evaluate the out-of-distribution (OOD) performance. Figure 4 presents results on the MedMCQA (Pal et al., 2022) and CS-Bench (Song et al., 2024) datasets. Although medical and computer science data account for only a negligible fraction of our training corpus (less than 1%), all methods show clear improvement after RL training. Among them, our RGR-GRPO achieves the best OOD performance, revealing its strong potential for generalization.

### 4.3. Analysis of Pass@*k* Curves Across Subjects

Pass@*k* measures the probability that at least one out of *k* independently sampled responses is correct. We use it to analyze how reinforcement learning (RL) expands the reachable reasoning space across different scientific domains. We

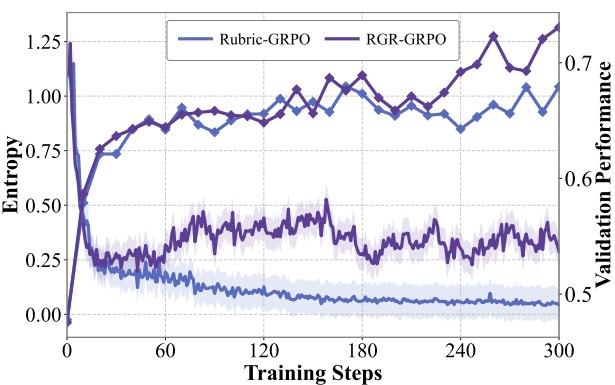

*Figure 6.* Comparison of entropy and validation between on-policy Rubric-GRPO and off-policy RGR-GRPO.

set the sampling temperature to 1.0 and evaluate *Qwen2.5-7B-Base* by sampling $k$ responses on the SciBench physics, chemistry, and math subsets. As shown in Figure 3, all baselines exhibit substantial improvements as $k$ increases, while RL-trained models already outperform the base model substantially at Pass@1. As $k$ grows, the performance gap gradually narrows, and the base model even surpasses the standard on-policy R1-GRPO baseline at $k=64$ on the Sci-Math subset. This phenomenon aligns with the observation of Cheng et al. (2025), suggesting that mathematical gains primarily stem from leveraging pre-trained knowledge rather than introducing fundamentally new reasoning abilities. Current on-policy RL formulations mainly sharpen the model's existing capabilities rather than expanding its reasoning horizon (Setlur et al., 2025; Yue et al., 2025; Wang et al., 2026).

In contrast, our RGR-GRPO breaks this limitation by incorporating reliable off-policy guidance through rubric-based supervision. It achieves consistently superior Pass@$k$ performance across all subjects, and its improvement persists more steadily as $k$ increases, demonstrating a stronger ability to promote effective policy exploration and reasoning diversity.

### 4.4. Policy Exploration in GRPO with Rubric Guidance

We further analyze the policy exploration dynamics during GRPO training for our RGR-GRPO method. As shown in Figure 5, the proportion of zero-reward responses decreases steadily as training progresses, while the mixture ratio of off-policy data gradually declines. This indicates that, in *Step1: Exploration Assessment*, on-policy responses increasingly contribute to the best-performing samples. In addition, the importance-sampling probabilities of both on-policy and off-policy rollouts remain stable throughout training, suggesting that the policy-mixing strategy is well balanced and does not collapse.

Furthermore, Figure 6 presents the evolution of training entropy and validation performance. Compared with the on-policy Rubric-GRPO, which exhibits rapid entropy collapse and fast convergence, our RGR-GRPO shows a distinct entropy trajectory: it drops sharply in the early steps

| Ablation Setting | Average Score |
|---|---|
| Qwen2.5-7B-Base | 44.6 |
| + Rubric-GRPO (Fact-Only Rubrics) | 53.5 |
| + Rubric-GRPO (All Rubrics) | 53.7 |
| + RGR-GRPO (w/o EA) | 53.8 |
| + RGR-GRPO (w/o Shaping) | 54.5 |
| + RGR-GRPO (Fact-Only Rubrics) | 55.2 |
| **+ RGR-GRPO (Full)** | **55.8** |

*Table 2.* Ablation results on average performance.

and then fluctuates between 0.2 and 0.4, indicating sustained and effective reasoning exploration. As the on-policy improvement saturates in the later stages, RGR-GRPO continues to enhance performance through off-policy rubric guidance, effectively breaking the exploration bottleneck.

### 4.5. Ablation Study

We conduct an ablation study on *Qwen2.5-7B* to analyze the impact of different rubric categories and the configurations of off-policy shaping and *exploration assessment (EA)*, as shown in Table 2. The results demonstrate the essential contribution of each component to the overall performance.

## 5. Exploring the Roles of Rubrics in Reward Enrichment and Self-Refinement

In this section, we analyze the roles of rubrics in reward provision and self-refinement in isolation within the dynamic RL training process, which also provides a clear motivation for our RGR-GRPO framework.

### 5.1. Rubric-Based Dense and Effective Rewards

To further evaluate the impact of rubric-based rewards in our RGR-GRPO framework, we compare GRPO under three reward mechanisms during online training, using the setup from Section 3. Figure 7 illustrates the dynamic training behavior of *Qwen2.5-7B-Base* under these rewards. Compared with the others, the Outcome reward yields a notably lower average reward due to its sparse 0–1 feedback. The large proportion of zero rewards indicates that most sampled groups lack informative signals, preventing the model from effective learning. Although the Likert reward provides dense feedback, it tends to be noisy and overly dependent on the judge model's preference, leading to unstable optimization and mediocre performance on MMLU-Pro (Wang et al., 2024). In contrast, the Rubric-based reward decomposes evaluation into explicit criteria, allowing the judge model to assign reliable binary decisions for each aspect and aggregate them into a dense and interpretable reward. This design enables dense and effective reinforcement learning, resulting in steady performance improvement and the best outcomes on MMLU-Pro.

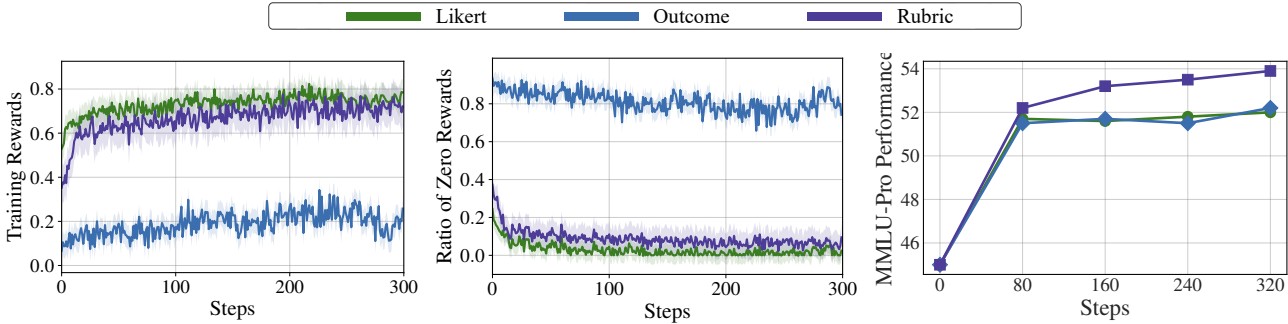

*Figure 7.* Performance dynamics of RL training under different reward settings.

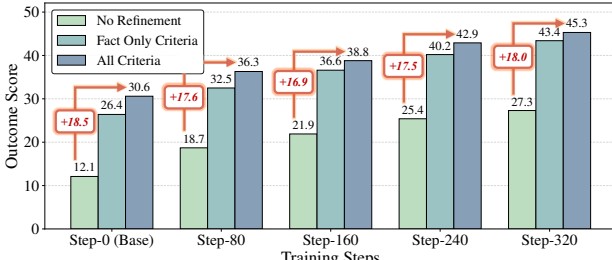

*Figure 8.* Performance improvements from rubric-based self-refinement across different RL training stages.

### 5.2. Rubric-Guided Self-Refinement at Test Time

Rubric-guided self-refinement can enhance the quality of model generations (Cook et al., 2024). To further explore its impact on complex reasoning in RL training, we evaluate checkpoints from the Outcome-GRPO training process described in Section 5.1. Specifically, we test the reasoning accuracy across different training steps using 4K additional multi-domain samples from *WebInstruct-Verify*, with Math-Verify used for evaluation. We compare three settings: **No Refinement**: direct generation without rubric guidance; **Fact-Only Criteria**: refinement prompted with factual rubrics only; **All Criteria**: refinement guided by both Factual and Process rubrics. For the latter two settings, we augment the model prompt with *"Keep in mind the following criteria:"* followed by the corresponding rubric instructions.

Experimental results in Figure 8 show consistent and substantial improvements with Rubric-Guided Self-Refinement. As training progresses and the model's inherent reasoning ability improves, the gain from rubric-based guidance remains stable rather than diminishing. This highlights the sustained benefits of rubrics as offline guidance throughout the reinforcement learning process: they not only enhance reasoning robustness during policy rollouts but also show potential to improve exploration when the on-policy learning signal becomes saturated.

## 6. Related Work

**Rubric-based Approaches.** Rubric-based evaluation provides structured and interpretable supervision by verifying each criterion separately and aggregating the results (Arora

et al., 2025; Galvan-Sosa et al., 2025; Winata et al., 2025). Recent studies have used rubrics to compute fine-grained rewards for RL training (Team et al., 2025; Gunjal et al., 2025; Viswanathan et al., 2025), which are particularly useful for tasks without definitive ground-truth answers. Some recent efforts further integrate rubric-guided signals into the rollout process to enhance policy learning (Zhou et al., 2025; Jayalath et al., 2025). However, unconstrained rubric design in complex reasoning tasks can lead to reward hacking or conflicting objectives (Eisenstein et al., 2023; Fu et al., 2025). To ensure stable and verifiable supervision, we restrict rubrics to two orthogonal dimensions—*factual* and *process*—capturing both correctness and reasoning quality while preserving generality across domains.

**Off-policy Guided Exploration.** Reinforcement learning (RL) has demonstrated significant improvements across diverse domain tasks (Li et al., 2026; Tong et al., 2026). However, on-policy RL often suffers from limited exploration and entropy collapse (Wu et al., 2025). While recent solutions like prolonged training (Liu et al., 2025a), entropy-based regularization (Dong et al., 2025; Zheng et al., 2025) and external guidance (Zhang et al., 2025a; RRV et al., 2025) can partially mitigate these issues, their effects remain limited when the exploration boundary is inherently constrained by the on-policy distribution. Recent studies introduce off-policy guidance to expand exploration by leveraging external responses or heuristic rollouts (Yan et al., 2025; Zhang et al., 2025b; Zhou et al., 2025). However, in complex reasoning domains, unconstrained off-policy rollouts often lead to instability, such as entropy explosion or semantic drift. Our RGR-GRPO addresses these challenges through a combination of *exploration assessment*, rubric-constrained guidance, and adaptive length penalties, which jointly preserve stability and diversity—allowing RGR-GRPO to explore beyond the on-policy limit while maintaining robust learning dynamics.

## 7. Conclusion

In this work, we propose RGR-GRPO, a rule-driven reinforcement learning framework for multi-domain reasoning. RGR-GRPO employs fine-grained rewards

derived from rubrics composed of *factual* and *process* criteria, and performs self-refinement by analyzing failed criteria from the best on-policy rollouts—thereby breaking the exploration ceiling. Evaluated across 14 benchmarks spanning multiple domains, RGR-GRPO consistently outperforms existing RL baselines. Notably, it maintains stable entropy fluctuations during training and achieves superior pass@k performance, demonstrating both sustained exploration and effective bottleneck breakthroughs.

## Impact Statement

This paper presents work whose goal is to advance the field of Machine Learning, specifically by improving the reasoning capabilities and training stability of Large Language Models (LLMs) through rubric-based reinforcement learning. By introducing fine-grained supervision and promoting effective exploration, our method aims to enhance the reliability of LLMs in complex scientific domains such as mathematics, physics, and chemistry. While these advancements hold promise for accelerating scientific research and educational applications, we acknowledge the general societal implications associated with the development of increasingly capable reasoning models. We believe that our approach, which emphasizes verifiable process criteria, contributes positively towards the development of more transparent and aligned AI systems. There are no other specific negative societal consequences we feel must be highlighted here.

## Acknowledgments

We thank Professor Sherry Tongshuang Wu for her valuable guidance and insightful feedback throughout the development of this work. This work is supported in part by the National Key R&D Program of China under Grant Nos. 2025YFC3309700, Beijing Natural Science Foundation No. 4262033, and the National Natural Science Foundation of China under Grant Nos. U25B2076, 62441229, and 62377043.

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

| Subject | Mathematics | Physics | Chemistry | Business | Finance | Economics | History | Biology | Psychology | Health | CS. | Other |
|---|---|---|---|---|---|---|---|---|---|---|---|---|
| **Count** | 3529 | 2375 | 1353 | 993 | 623 | 556 | 272 | 263 | 75 | 52 | 48 | 435 |
| **Ratio** | 33.5% | 22.6% | 12.9% | 9.4% | 5.9% | 5.3% | 2.6% | 2.5% | 0.7% | 0.5% | 0.5% | 4.1% |

*Table 4.* Distribution of subjects in the training and validation dataset.

## A. Experimental Details

### A.1. Datasets

We use the *WebInstruct-Verify* (Ma et al., 2025) dataset to train GRPO. *WebInstruct-Verify* covers multiple disciplines and provides verified ground-truth answers, which allows fair comparison with outcome-based reward baselines. We randomly sample subsets and used *Qwen2.5-7B-Base* to generate initial responses, then filtered out simple questions using Likert scores. For each prompt we produce a reference answer with *O3* and remove prompts whose reference answers were excessively long. This yielded about 10k high-quality, multi-domain training examples; the subject distribution is reported in Table 4, with the majority concentrated in mathematics, physics,

| Rubric Type | Count | Ratio |
|---|---|---|
| Factual | 13,809 | 27.8% |
| Process | 35,926 | 72.2% |

*Table 3.* Distribution of rubric types across all training and validation examples.

and chemistry. Although Computer Science and Health constitute only small fractions of the data, they nevertheless perform well in their respective evaluations (see Section 4.2). From each prompt and its reference answer we derive two rubric types (Factual and Process) and their counts are summarized in Table 3.

### A.2. Training

We train GRPO on *Qwen2.5-Base* using the *Verl* [2] framework, which employs *vLLM* [3] as the rollout generator. The detailed hyperparameter settings are provided in Table 5 for both on-policy methods (outcome-, Likert-, and rubric-GRPO) and off-policy methods (Critique-GRPO (Zhang et al., 2025b), LUFFY (Yan et al., 2025), and our RGR-GRPO). Specifically, training is conducted on 8 H100 GPUs, while the LLM-as-Judge reward service is deployed across multiple machines equipped with 4 L40S GPUs each.

### A.3. Evaluation

We follow the *Language Model Open Science Evaluation* framework[4] framework for all tasks evaluation (Fan et al., 2025), an open-source evaluation suite designed for standardized and reproducible benchmarking of large language models. This system supports both conversation and base models, provides flexible integration of new tasks, and enables large-scale multi-node evaluations with detailed instance-level outputs.

To comprehensively assess reasoning ability, we adopt the same benchmark collection as in the *Open Science Evaluation Suite*, covering a wide range of disciplines and reasoning types. Specifically, we evaluate models on:

- **General Scientific Reasoning:** MMLU (Hendrycks et al., 2020), MMLU-Pro (Wang et al., 2024), GPQA-Diamond (Rein et al., 2023), GPQA-Main (Rein et al., 2023), and OlympicArena (Huang et al., 2024);

- **Mathematical Reasoning:** Sci-Math (Wang et al., 2023), MATH (Hendrycks et al., 2021), and MATH500 (Lightman et al., 2023).

- **Physics Reasoning:** PIQA (Bisk et al., 2020) and Sci-Physics (Wang et al., 2023);

- **Chemistry Reasoning:** ChemBench (Mirza et al., 2024), and Sci-Chemistry (Wang et al., 2023);

- **Out-of-distribution Reasoning:** CS-Bench (Song et al., 2024) and MedMCQA (Pal et al., 2022);

---

[2]https://github.com/volcengine/verl
[3]https://github.com/vllm-project/vllm
[4]https://github.com/GAIR-NLP/lm-open-science-evaluation

| Name | Value (for on-policy methods) | Value (for off-policy methods) | Description |
|---|---|---|---|
| num_training_prompts | 10k | 10k | Default number of prompts used for RL finetuning, unless otherwise specified. |
| training_steps | 400 | 400 | Total number of gradient update steps. |
| eval_freq | 40 | 40 | Interval (in updates) between two evaluations. |
| training_batch | 96 | 96 | Effective batch size accumulated per update. |
| learning_rate | $1e^{-6}$ | $1e^{-6}$ | Step size of optimizer during training. |
| max_prompt_length | 1024 | 1024 | Maximum length of input prompt tokens. |
| max_response_length | 5120 | 5120 | Maximum number of tokens for model outputs. |
| n_rollouts | 8 | 8 | Number of sampled rollouts for each prompt. |
| n_refine | 0 | 1 | Maximum number of off-policy refinements per rollout. |
| reward_range | [0,1] | [0,1] | Value range of scalar rewards. |
| kl_loss_coef | 0.001 | 0.0 | Weight assigned to KL divergence regularization. |
| $\gamma$ | None | 0.1 | Coefficient for policy shaping during off-policy updates. |
| train_temp | 1.0 | 1.0 | Sampling temperature used for training rollouts. |
| val_temp | 0.0 | 0.0 | Sampling temperature used in validation runs. |
| total_epochs | 4 | 4 | Number of complete passes over the dataset. |
| **Evaluation** | | | |
| eval_temp | 0.0 | 0.0 | Sampling temperature for evaluation generation. |
| max_tokens | 8192 | 8192 | Token budget limit for evaluation inference. |

*Table 5.* Default hyperparameters and configurations used in RL finetuning and evaluation.

This unified evaluation protocol allows fair and consistent comparison across diverse scientific domains. We set the generation temperature to 0.0 and the maximum response length to 8192 for all evaluation tasks.

| Category | Benchmark | Question Type | CoT | Unit | Metric |
|---|---|---|---|---|---|
| General Reasoning | MMLU (Hendrycks et al., 2020) | Multi-Choice | ✓ | ✗ | EM |
| | MMLU-Pro (Wang et al., 2024) | Multi-Choice | ✓ | ✗ | EM |
| | GPQA-Diamond (Rein et al., 2023) | Multi-Choice | ✓ | ✗ | EM |
| | GPQA-Main (Rein et al., 2023) | Multi-Choice | ✓ | ✗ | EM |
| | OlympicArena (Huang et al., 2024) | Computational Problems | ✓ | ✓ | EM (unit) |
| Math | Sci-Math (Wang et al., 2023) | Computational Problems | ✓ | ✓ | EM |
| | MATH (Hendrycks et al., 2021) | Computational Problems | ✓ | ✗ | EM |
| | MATH500 (Lightman et al., 2023) | Computational Problems | ✓ | ✗ | EM |
| Chemistry | ChemBench (Mirza et al., 2024) | Multi-Choice & Problem-Solving | ✓ | ✗ | EM |
| | Sci-Chemistry (Wang et al., 2023) | Computational Problems | ✓ | ✓ | EM |
| Physics | PIQA (Bisk et al., 2020) | Multi-Choice | ✓ | ✗ | EM |
| | Sci-Physics (Wang et al., 2023) | Computational Problems | ✓ | ✓ | EM |
| Computer Science | CS-Bench (Song et al., 2024) | Multi-Choice & True/False | ✓ | ✗ | EM |
| Medicine | MedMCQA (Pal et al., 2022) | Multi-Choice | ✓ | ✗ | EM |

*Table 6.* The evaluation configurations used in our experiments. **CoT** denotes evaluations conducted with chain-of-thought prompting. **Unit** indicates that the answer requires a correct physical unit. **EM (unit)** measures exact match accuracy considering both the numerical value and its associated unit.

# B. RGR-GRPO Algorithm

We summarize the main algorithmic pipeline of our RGR-GRPO in Algorithm 1. It consists of three stages: first, we assess whether off-policy guidance is needed to overcome exploration limitations (Exploration Assessment); then, we refine the best rollout based on failed rubrics (Rubric-Based Self-Refinement); finally, we merge the on-policy and off-policy rollouts to update the policy model (Mix-Policy GRPO).

**Algorithm 1** RGR-GRPO: Reward and Guidance through Rubrics

**Input:** Pretrained LLM policy $\pi_{\text{old}}$ parameterized by $\theta$, reward model $\pi_{RM}$, rubric-based reward function $\text{Reward}(\cdot)$, question set $Q = \{q\}$, and self-refinement template $T_{\text{refine}}$

**Goal:** Incorporate rubric-based reward and off-policy refinement to improve reasoning robustness

**for** each question $q \in Q$ **do**

    **Step 1: Exploration Assessment**

    Sample $G-1$ initial responses from the old policy:

$$\{o_i\}_{i=1}^{G-1} \sim \pi_{\text{old}}(\cdot \mid q)$$

    Evaluate each response with the rubric-based reward function:

$$r_i, \; \{s_k(q,o_i)\}_{k=1}^{|\mathcal{C}_i|} \leftarrow \text{Reward}(q,o_i)$$

    Identify the best-performing response:

$$o_{\text{best}} = \underset{o_i \in \{o_1,\ldots,o_{G-1}\}}{\text{argmax}} r_i$$

    **if** $\sum_{k=1}^{|\mathcal{C}_{\text{best}}|} s_k(q,o_{\text{best}}) = |\mathcal{C}_{\text{best}}|$ **then**

        On-policy exploration suffices. Perform on-policy GRPO update using on-policy rollouts:

$$o_G \sim \pi_{\text{old}}(\cdot \mid q), \quad \mathcal{J}_{\text{On-policy}} = \mathbb{E}_{q \sim \mathcal{D}, \{o_i\}_{i=1}^{G} \sim \pi_{\theta_{\text{old}}}(\cdot \mid q)} \frac{1}{G} \sum_{i=1}^{G} \frac{1}{|o_i|} \sum_{t=1}^{|o_i|} r_t^{(i)}(\theta) \hat{A}_i$$

        where

$$r_G \leftarrow \text{Reward}(q,o_G), \quad r_t^{(i)}(\theta) = \frac{\pi_\theta(o_i \mid q, o_{<t}^{(i)})}{\pi_{\theta_{\text{old}}}(o_i \mid q, o_{<t}^{(i)})}, \quad \hat{A}_i = \frac{r_i - \text{mean}(\{r_j\}_{j=1}^{G})}{\text{std}(\{r_j\}_{j=1}^{G})},$$

        **Continue to next iteration**

    **end if**

    **Step 2: Rubric-Based Self-Refinement**

    Identify failed rubric items:

$$\mathcal{C}^{\text{failed}} = \{c_k \mid s_k(q,o_{\text{best}}) = 0\}$$

    Generate an off-policy refined response conditioned on the failures:

$$o_G \sim \pi_{\text{old}}(\cdot \mid q, o_{\text{best}}, \mathcal{C}^{\text{failed}}), \quad r_G \leftarrow \text{Reward}(q,o_G)$$

    **Step 3: Mix-Policy GRPO**

    Shape the off-policy probability ratios:

$$f_{\text{shape}}(r_t^{(G)}(\theta)) = \frac{\pi_\theta(o_t^{(G)} \mid q, o_{<t}^{(G)})}{\pi_\theta(o_t^{(G)} \mid q, o_{<t}^{(G)}) + \gamma}$$

    Optimize the mixed-policy objective combining on- and off-policy terms:

$$\mathcal{J}_{\text{RGR-GRPO}}(\theta) = \mathbb{E}_{q \sim Q, \{o_i\}_{i=1}^{G-1} \sim \pi_{\text{old}}, \; o_G \sim \pi_{\text{old}}(\cdot \mid q, o_{\text{best}}, \mathcal{C}^{\text{failed}})}$$

$$\frac{1}{G} \left[ \underbrace{\sum_{i=1}^{G-1} \sum_{t=1}^{|o_i|} r_t^{(i)}(\theta) \hat{A}_i}_{\text{On-policy objective}} + \underbrace{\sum_{t=1}^{|o_G|} f_{\text{shape}}(r_t^{(G)}(\theta)) \hat{A}_G}_{\text{Off-policy objective}} \right]. \tag{12}$$

    **Output:** Fine-tuned LLM policy $\pi_\theta$

**end for**

## C. Necessity of Exploration Assessment

**Theorem C.1** (Necessity of Exploration Assessment). *The Exploration Assessment (EA) mechanism, which conditionally applies off-policy refinement (Section 2.3, Step 2-3) only when the on-policy exploration upper bound is insufficient (i.e., no perfect solution $o_{best}$ is found), is a necessary component for stabilizing the RGR-GRPO training process. It functions as an adaptive variance controller, preventing unnecessary, high-variance gradient updates that can lead to excessive distributional shift and training collapse (i.e., entropy explosion).*

*Proof.* Our proof is structured by analyzing the variance and distributional impact of the two distinct gradient estimators employed by the RGR-GRPO framework, contingent on the EA's decision.

**1. Definitions: Gradient Estimators and Distributions** Let us define the two relevant sampling distributions:

- **On-policy distribution** $\pi_{\mathrm{on}}$: This is the baseline policy from which initial rollouts are sampled, $\pi_{\mathrm{on}} = \pi_{\theta_{\mathrm{old}}}(\cdot \mid q)$.

- **Off-policy refinement distribution** $\pi_{\mathrm{refine}}$: This is the conditional policy used to generate the refined response $o_G$, $\pi_{\mathrm{refine}} = \pi_{\theta_{\mathrm{old}}}(\cdot \mid q, o_{\mathrm{best}}, \mathcal{C}^{\mathrm{failed}})$.

When $\mathcal{C}^{\mathrm{failed}}$ is non-empty (i.e., refinement is needed), $\pi_{\mathrm{refine}}$ is explicitly conditioned on new information, making it distinct from the base policy. Thus, the Kullback-Leibler (KL) divergence is non-zero: $D_{KL}(\pi_{\mathrm{refine}} \| \pi_{\mathrm{on}}) > 0$.

Based on the EA's decision, one of two gradient estimators is used for the policy update $\theta_{t+1} \leftarrow \theta_t + \eta \mathbf{g}$:

- **On-Policy Gradient (EA "If" Branch):** $\mathbf{g}_{\mathrm{on}}$. This is the standard on-policy GRPO gradient (as in Eq. (1 -2)), where all $G$ samples, including $o_G$, are drawn from $\pi_{\mathrm{on}}$:

$$\mathbf{g}_{\mathrm{on}} = \nabla_\theta \mathbb{E}_{\{o_i\}_{i=1}^G \sim \pi_{\mathrm{on}}} \left[ \frac{1}{G} \sum_{i=1}^G \sum_{t=1}^{|o_i|} r_t^{(i)}(\theta) \hat{A}_i \right] \tag{13}$$

- **Mix-Policy Gradient (EA "Else" Branch):** $\mathbf{g}_{\mathrm{mix}}$. This is the mixed-policy objective from Eq. (10), based on $G-1$ samples from $\pi_{\mathrm{on}}$ and one sample $o_G$ from $\pi_{\mathrm{refine}}$:

$$\mathbf{g}_{\mathrm{mix}} = \nabla_\theta \mathbb{E}_{\substack{\{o_i\}_{i=1}^{G-1} \sim \pi_{\mathrm{on}} \\ o_G \sim \pi_{\mathrm{refine}}}} [\mathcal{J}_{\mathrm{RGR\text{-}GRPO}}(\theta)] \tag{14}$$

**2. Variance Analysis of Gradient Estimators** In reinforcement learning, training stability is inversely related to the variance of the policy gradient estimator, $\mathrm{Var}(\mathbf{g})$. High-variance gradients can cause large, erratic policy updates, leading to the "entropy explosion" or "policy collapse" phenomenon (Yan et al., 2025; Zhou et al., 2025).

- *Variance of $\mathbf{g}_{on}$:* $\mathrm{Var}(\mathbf{g}_{\mathrm{on}})$ represents the baseline, on-policy gradient variance, which is generally considered the most stable estimator.

- *Variance of $\mathbf{g}_{mix}$:* The estimator $\mathbf{g}_{\mathrm{mix}}$ is a sum of low-variance on-policy terms and one high-variance off-policy term:

$$\mathbf{g}_{\mathrm{mix}} = \frac{1}{G} \left[ \sum_{i=1}^{G-1} \mathbf{g}_{\mathrm{on}}^{(i)} + \mathbf{g}_{\mathrm{off}}^{(G)} \right] \tag{15}$$

where $\mathbf{g}_{\mathrm{off}}^{(G)}$ is the gradient of the off-policy objective $\mathcal{L}_{\mathrm{off}}^{(G)} = \sum_{t=1}^{|o_G|} f_{shape}(r_t^{(G)}(\theta)) \hat{A}_G$.

The off-policy gradient $\mathbf{g}_{\mathrm{off}}^{(G)}$ has an inherently high variance. This is because the loss function $\mathcal{L}_{\mathrm{off}}^{(G)}$ uses samples $o_G \sim \pi_{\mathrm{refine}}$ but relies on an importance sampling (IS) ratio $r_t^{(G)}(\theta)$ (or a function $f_{shape}$ of it) whose denominator is based on $\pi_{\mathrm{on}} = \pi_{\theta_{\mathrm{old}}}$ (as defined in Eq. (10)). The variance of IS-based estimators scales with the divergence between the sampling distribution and the target distribution (Shapiro, 2003). In this case, the mismatch $D_{KL}(\pi_{\mathrm{refine}} \| \pi_{\mathrm{on}}) > 0$ fundamentally leads to $\mathrm{Var}(\mathbf{g}_{\mathrm{off}}^{(G)}) \gg \mathrm{Var}(\mathbf{g}_{\mathrm{on}}^{(i)})$.

Consequently, the overall variance of the mixed-policy gradient is significantly higher than the pure on-policy gradient:

$$\mathrm{Var}(\mathbf{g}_{\mathrm{mix}}) \approx \frac{(G-1)\mathrm{Var}(\mathbf{g}_{\mathrm{on}}^{(i)}) + \mathrm{Var}(\mathbf{g}_{\mathrm{off}}^{(G)})}{G^2} > \frac{G \cdot \mathrm{Var}(\mathbf{g}_{\mathrm{on}}^{(i)})}{G^2} \approx \mathrm{Var}(\mathbf{g}_{\mathrm{on}}) \tag{16}$$

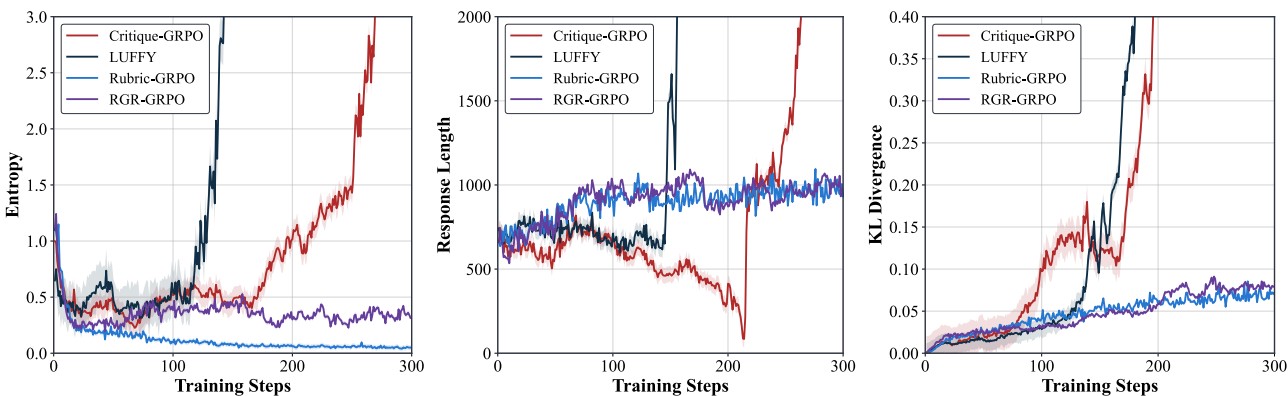

*Figure 9.* Training dynamics of policy entropy, response length, and KL divergence for mix-policy RL methods.

**3. The Role of Exploration Assessment as an Adaptive Controller**   The EA mechanism dynamically selects between these two estimators based on the policy's current performance, effectively balancing exploration efficacy with update stability.

- **Case 1: Sufficient Exploration ($\sum s_k = |\mathcal{C}_i|$).** The on-policy exploration upper bound is sufficient; $\pi_{\text{on}}$ can already generate an optimal solution $o_{\text{best}}$.

    - *EA Action:* The EA selects the **low-variance** update $\mathbf{g}_{\text{on}}$.
    - *Justification:* The goal of exploration (finding a perfect solution) is already met. The optimal action is to reinforce this existing behavior stably.
    - *Risk Avoided:* If we were to *unconditionally* use $\mathbf{g}_{\text{mix}}$ (i.e., without EA), we would be injecting a high-variance $\mathbf{g}_{\text{off}}^{(G)}$ gradient *unnecessarily*. This high-variance, "risky" update provides no additional benefit (the policy is already optimal) but introduces a significant risk of destabilizing the policy, i.e., $D_{KL}(\pi_{\theta_{t+1}} \| \pi_{\theta_t})$ could be large and uncontrolled.

- **Case 2: Insufficient Exploration ($\sum s_k < |\mathcal{C}_i|$).** The on-policy exploration upper bound is low; $\pi_{\text{on}}$ is stuck in a sub-optimal space, and $o_{\text{best}}$ is imperfect.

    - *EA Action:* The EA selects the **high-variance** update $\mathbf{g}_{\text{mix}}$.
    - *Justification:* The stable, low-variance update $\mathbf{g}_{\text{on}}$ would merely reinforce the sub-optimal $o_{\text{best}}$. The high-variance $\mathbf{g}_{\text{off}}^{(G)}$ term, while risky, is *necessary*. It forces the policy to learn from the new, high-reward, out-of-distribution trajectory $o_G \sim \pi_{\text{refine}}$, effectively expanding the exploration boundary. This is a deliberate "stability-for-exploration" trade-off.

**Conclusion.**   The Exploration Assessment acts as an adaptive variance controller. It defaults to the stable, low-variance on-policy update $\mathbf{g}_{\text{on}}$ whenever possible (Case 1), thus preserving training stability. It only engages the high-variance, exploratory $\mathbf{g}_{\text{mix}}$ update when it is strictly necessary to overcome the limitations of on-policy exploration (Case 2).

By preventing the *unnecessary* application of high-variance, off-policy gradient corrections, the Exploration Assessment mechanism minimizes the average gradient variance over the training horizon, thereby substantially reducing the risk of entropy explosion and ensuring a more robust and stable policy optimization process. □

## D. Analysis of Off-Policy Training Stability

Despite their reported best performance during training, our reproductions of LUFFY (Yan et al., 2025) and Critique-GRPO (Zhang et al., 2025b) performed worse than anticipated in Table 1, frequently underperforming their on-policy counterparts. We attribute this discrepancy primarily to the adverse effects of excessive distributional shift introduced by off-policy guidance. Figure 9, which plots key training metrics including policy entropy, response length, and KL divergence, reveals this instability of these off-policy baselines. Both Critique-GRPO and LUFFY experience training collapse mid-run

(around 100+ steps), characterized by an uncontrolled spike in all three metrics. Qualitative inspection of the generated outputs at this stage confirmed this collapse, highlighting the inherent instability of these off-policy methods.

This observation aligns with the findings of Zhou et al. (2025), who attempted to mitigate this issue by using a sigmoid function scheduled over training steps to control the off-policy mixing ratio, effectively reverting to purely on-policy rollouts in the latter half of training. However, such a time-based schedule lacks robustness as it is not adaptive to the policy's actual learning progress. In contrast, our RGR-GRPO employs the *Exploration Assessment* mechanism to dynamically control the mixing ratio of off-policy rollouts throughout the entire training process. As shown in Figure 5 (Left), this ratio adaptively and smoothly decreases as the policy's actual exploration boundary improves. Further evidence of this stability is presented in Figure 9, where RGR-GRPO's curves for entropy, response length, and KL divergence are nearly identical to those of the stable on-policy Rubric-GRPO, confirming its robustness.

## E. Prompt Templates

We provide all prompt templates used in our experiments. During RL training, we deploy *Qwen3-32B* as the LLM-as-Judge to supply reward feedback for the Likert- and rubric-based methods (see Section E.1). In our RGR-GRPO, the prompt-policy model listed in Section E.2 is used to produce refined responses conditioned on failed rubrics. For every filtered example, we use *O3* to generate question-specific rubrics according to Section E.3.

### E.1. LLM-Judge Prompts

---

**Prompt for Likert-Reward Judge**

**System Prompt:**
You are an expert evaluator. Given a user prompt, a reference response, and a generated response, please rate the overall quality of the generated response on a scale of 1 to 10 based on how well it compares to the reference response. Consider factors such as accuracy, completeness, coherence, and helpfulness when comparing to the reference. The reference response represents a high-quality answer that you should use as a benchmark. Start your response with a valid JSON object that starts with "```json" and ends with "```". The JSON object should contain a single key "rating" and the value should be an integer between 1 and 10.
Example response:
```json
{
"rating": 7
}```

**User Prompt:**
Given the following prompt, reference response, and generated response, please rate the overall quality of the generated response on a scale of 1 to 10 based on how well it compares to the reference.

```
<prompt>
{prompt}
</prompt>

<reference_response>
{reference}
</reference_response>

<generated_response>
{response}
</generated_response>
```

Your JSON Evaluation:

---

---

**Prompt for Rubric-Reward Judge**

**System Prompt:**
You are an expert evaluator. Given a user prompt, a generated response, and a single quality criterion, please judge whether the response satisfies the criterion (1) or does not satisfy the criterion (0). Do not consider any other quality aspects outside the provided criterion. Your evaluation must be strictly limited to whether the response meets the specified criterion. Start your response with a valid JSON object that starts with "```json" and ends with "```". The JSON object should contain a single key "rating" and the value should be either 1 (criterion satisfied) or 0 (criterion not satisfied).
Example response:
```json
{
"rating": 1
}```

**User Prompt:**
Given the following prompt, response, and evaluation criterion, please judge whether the response satisfies the specified criterion (1) or does not satisfy it (0). Ignore all other factors outside the criterion.

```
<prompt>
{prompt}
</prompt>

<response>
{response}
</response>

<criterion>
{single_rubric_criterion}
</criterion>
```

Your JSON Evaluation:

---

### E.2. Self-Refinement with Failed Rubrics

**Prompt for Rubric-Guided Refinement**

**System:** You are a helpful assistant.
**User:** Given the following inputs:
Question:

```
{base_text}
```

Previous Response:

```
{selected_response}
```

Rubrics:

```
{rubrics}
```

Instruction:
Refine the Previous response so it fully satisfies all items in Rubrics. Provide clear, concise, step-by-step reasoning that leads to the correct result. Put your final answer within \boxed{}.
Refined Response:

---

## E.3. Rubric Generation

---

**Prompt for Rubric Generator**

**System Prompt:**
You are an expert rubric writer for a wide range of academic disciplines (e.g., Physics, Chemistry, Mathematics). Your job is to generate a self-contained set of evaluation criteria ("rubrics") for judging how good a response is to a given question in one of these disciplines. Rubrics may cover aspects such as factual correctness, reasoning/process, completeness, and computational accuracy. Each rubric item must be fully self-contained so that a non-expert reader can easily and unambiguously decide whether the criterion is satisfied.

**Inputs:**

- `question`: The full question text.

- `reference_answer`: The ideal (model) answer, including any key facts, explanations and reasoning steps.

- `groundtruth`: The correct final fact(s) or result(s) for the question.

**High-level Rules:**

- **Total items:** Generate between 3 and 10 rubric items, choosing the number appropriate to the complexity of the question and prioritizing the most important evaluation aspects for the discipline and task.

- **Categories:** Use exactly two category types and these exact prefix strings at the start of every `description`:

    - **Factual Criteria:** for verifiable final facts or answers.
    - **Process Criteria:** for key intermediate steps, calculations, theorems, or reasoning required to derive the final answer.

- **When to generate each category:**

    - **Factual Criteria:** Generate only when the question has one or more determinable, verifiable final facts or answers. If the question has multiple independent final facts or parts, generate a separate Factual Criterion for each. The response is factually correct only if all Factual Criteria are satisfied.
    - **Process Criteria:** Generate for the *necessary* intermediate results, explicit sub-answers, key equations, definitions, or theorems that must be invoked. Each Process Criterion must be concrete and testable (e.g., "States that the derivative of $\sin(x)$ is $\cos(x)$" or "Shows substitution $u = x^2$ in the integral"). Avoid vague wording such as "uses proper reasoning."

**Item Format Rules:**

- Each rubric item must be an object with exactly two keys: `description`, `weight`.

- `description`: One sentence beginning with the category prefix, followed by a clear, testable statement.

- `weight`: Integer from 1 to 5 (5 = most important).

- Description examples:

    - Factual Criteria: States the correct final value of the integral as $\pi/2$.
    - Factual Criteria: Identifies mitochondria as the site of ATP production in eukaryotic cells.
    - Process Criteria: Shows the substitution $u = x^2$ and adjusts the limits accordingly before integrating.
    - Process Criteria: Applies Newton's second law F = ma to set up the differential equation for motion.

**Additional Requirements:**

- All rubric items must be easy to evaluate as satisfied or not satisfied.

- Avoid vague criteria like "demonstrates understanding" or "explains clearly."

- Output **only** the JSON array of rubric objects, each with keys `description` and `weight`.

- The `description` must begin with one of the two exact prefixes above.

- Each rubric item must evaluate a distinct, non-overlapping aspect of the answer. No duplication across items.

**User Prompt:**
Now, given the question, reference answer, and groundtruth, generate the rubric as described. The reference answer is an ideal response but may not be fully correct or exhaustive; use it only as a guide.

---

