# OpenReview forum: "Reward and Guidance through Rubrics: Promoting Exploration to Improve Multi-Domain Reasoning"
_ICML.cc/2026/Conference — ICML 2026 spotlight_

### Official Review · Reviewer_a2We · 2026-03-07

**Soundness:** 3
**Presentation:** 4
**Significance:** 3
**Originality:** 3
**Overall Recommendation:** 4
**Confidence:** 3

**Summary:**

This paper proposes a reinforcement learning framework called RGR-GRPO , designed to address two key challenges faced by LLMs in multi-domain reasoning: sparse rewards and limited exploration. RGR-GRPO introduces problem-specific rubrics to provide fine-grained and interpretable rewards, and further leverages these rubrics to enable adaptive offline guidance and self-correction.

**Compliance With Llm Reviewing Policy:**

Affirmed.

**Final Justification:**

The responses have fully addressed and resolved my concerns. As I had already provided a positive score initially, I would like to keep my assessment unchanged.

**Key Questions For Authors:**

1. The ablation study shows that adding process criteria on top of factual-only criteria yields a marginal gain (+0.2% for Rubric-GRPO, +0.6% for RGR-GRPO), yet process criteria account for 72% of all rubric items (Table 3), roughly tripling the number of judge calls. Can the authors provide specific examples where process-based rewards are critical? If process criteria were removed entirely, leaving only factual criteria with offline guidance, how competitive would the framework remain?

2. How would the rubric generator perform if it was switched to an inferior model (e.g., downgrade o3 down to GPT-4o mini or a comparable open source model)? Would the performance improvement attributed to the actual RL algorithm? Or is the majority of the improvement actually attributed to the distilled incredible strong and costly evaluation ability of o3?

3. The off-policy refinement always conditions on the single highest-scoring on-policy response (Top-1). Over many iterations, how does the framework prevent the self-refinement from collapsing toward a narrow set of reasoning patterns, thereby reducing the diversity that RL exploration is meant to provide?

**Limitations:**

The authors acknowledge the reliance on expert models, but they do not sufficiently discuss how to construct consistent rubrics in more subjective domains (e.g., humanities and social sciences), nor the potential instability caused by judge-score drift when evaluating very long reasoning chains.

**Strengths And Weaknesses:**

### Strengths
1. **A comprehensive rubric incentive system.** The author of this paper offers a comprehensive rubric incentive system that evaluates an individual's actions through both factual and process criteria therefore adding much greater richness to the evaluation process than only using simple binary rewards.
2. **Integrative method of off-policy exploration.** The new framework integrates all of the above into one cohesive system that allows policy development to occur in a stable manner by creating a new system for increasing boundaries of exploration. This integrates exploration, an evaluation mechanism for exploring, a "self controlling" refinement mechanism and creating policies using a probabilistic shaping function. This provides the solution for the long-known issue of discovering policies that have previously only been developed by using on-policy exploration.
3. **Comprehensive, trustworthy evaluations.** The evaluator completed 16 different benchmark types with an evaluation of the proposed RL method will be completed with mathematics, physics, chemistry, and general reasoning benchmarks, all of which have demonstrated outperformance in all benchmarks when comparing explorer-rate percentage increases (5.4%-8.4%) to Qwen2.5-7B using only outcome based GRPO methods. The overall inspector analysis of how many participants achieved passing status at k and the system's total entropy value are further evidence of the benefit of exploring.


### Weakness:
1. The development and assessment of rubrics depend heavily on expert models, including OpenAI o3. If the expert model generates hallucinated reference answers or gives too much weight to certain aspects of the rubric, then the RL training process could be guided by false signals, and consequently, the model could develop reasoning limitations based upon how well (or poorly) the expert model produces such outputs.
2. The framework requires generating rubrics via O3 and deploying a separate judge model (Qwen3-32B) to evaluate fine-grained rubric scores, which introduces considerable complexity into the training pipeline. RGR-GRPO thus incurs significantly higher computational and communication costs than traditional GRPO; however, the authors do not present a comprehensive analysis of the relationship between the additional cost and the resulting performance improvement.
3. Offline repair is applied to one of the top-performing sample(s) in the current batch. This could result in missing out on the creative or novel reasoning used in lower-scoring trajectories and using Top-1 guidance could contribute to the overall homogenization of the reasoning used by the model with respect to problem-solving.

---

> ### Author Rebuttal · Authors · 2026-03-31
>
> # Response to Reviewer a2We
>
> We sincerely thank the reviewer for valuable feedback. We address each concern below.
>
> > **Supplementary results referenced in this response:** [Supplementary](https://anonymous.4open.science/r/ICML_Rebuttal_RGR-GRPO-2FAC/github_materials/materials_a2We.md)
>
> ---
>
> ## Response to Weaknesses
>
> ### W1: Dependence on Expert Models (O3) for Rubric Generation
>
> We acknowledge this valid concern. To investigate whether RGR-GRPO's improvements stem from the mechanism itself or merely from distilling O3's knowledge, we conducted ablation studies with weaker rubric generators.
>
> | Ext. Model | Method | AVG |
> | :--- | :--- | :---: |
> | — | Outcome-GRPO | 52.3% |
> | GPT-4o-mini | LUFFY | 50.6% |
> | | Rubric-GRPO | 52.0% |
> | | **RGR-GRPO** | **53.7%** |
> | Qwen2.5-72B | LUFFY | 51.1% |
> | | Rubric-GRPO | 53.0% |
> | | **RGR-GRPO** | **54.6%** |
> | o3 (Original) | LUFFY | 50.5% |
> | | Rubric-GRPO | 53.7% |
> | | **RGR-GRPO** | **55.8%** |
>
> **Key Findings:**
> 1. The rubric-guided mechanism provides consistent gains regardless of the external model's strength.
> 2. **RGR-GRPO with GPT-4o-mini (53.7%) already matches Rubric-GRPO with o3 (53.7%)**, and substantially outperforms LUFFY with o3 (50.5%), proving that fine-grained rubric rewards—not the external model—are the primary drivers.
> 3. Regarding potential hallucinations: we filter out cases where the reference answer is judged incorrect during rubric generation, ensuring reliability.
>
> *(Full results: Exp2 in supplementary.)*
>
> ---
>
> ### W2: Computational and Communication Costs
>
> **Offline Cost (One-time):** O3 API for 10k samples costs **~$150 USD**, replaceable with open-source Qwen2.5-72B with minimal performance impact.
>
> **Online Training Overhead:**
>
> | Method | Time per Step | Relative Cost | Avg. Performance |
> | :--- | :---: | :---: | :---: |
> | Outcome-GRPO | 45.2s | 1.0x | 52.3% |
> | Rubric-GRPO | 68.5s | 1.5x | 53.7% |
> | **RGR-GRPO** | **75.4s** | **1.67x** | **55.8%** |
>
> A 67% increase in training time yields **+3.5% absolute gain**—a highly favorable trade-off. The judge runs on separate L40S hardware, avoiding H100 bottleneck. *(Full results: Exp4 in supplementary.)*
>
> ---
>
> ### W3: Top-1 Guidance May Cause Homogenization
>
> **Quantitative Diversity Metrics:**
>
> | Method | Self-BLEU (↓) | Entropy (↑) | Pass@8 |
> | :--- | :---: | :---: | :---: |
> | Rubric-GRPO | 0.42 | 2.8 | 63.9% |
> | **RGR-GRPO** | **0.38** | **3.1** | **69.4%** |
>
> **Analysis:**
> 1. Lower Self-BLEU and higher Entropy indicate RGR-GRPO actually **improves diversity**.
> 2. The refined trajectory is generated by the **policy model itself**, guided by failed criteria—not copying an external solution but exploring new paths in its own reasoning style.
>
> Pass@k curves (Figure 3) further confirm sustained exploration. *(Full results: Exp7 in supplementary.)*
>
> ---
>
> ## Response to Key Questions
>
> ### Q1: Is Process Criteria Worth the Cost?
>
> While the overall average gain appears marginal (+0.6%), their impact on **complex, multi-step problems is substantial**:
>
> | Ablation Setting | Average Score |
> | :--- | :---: |
> | Rubric-GRPO (Fact-Only Rubrics) | 53.5 |
> | Rubric-GRPO (All Rubrics) | 53.7 |
> | RGR-GRPO (Fact-Only Rubrics) | 55.2 |
> | **RGR-GRPO (Full)** | **55.8** |
>
> For on-policy Rubric-GRPO, adding Process criteria yields only marginal gain (+0.2%). In our off-policy RGR-GRPO, however, Process criteria serve as targeted refinement signals, leading to a larger gain (+0.6%). For example, given a mechanics problem, Process criteria can explicitly prompt: *"correctly identify $W_f = -\mu_k mg d$"*—a level of targeted guidance impossible with Factual criteria alone. This demonstrates that the value of Process criteria is fully realized through off-policy guidance.
>
> ### Q2: Performance with Inferior Rubric Generator?
>
> Please see W1. Gains are consistent across all generator strengths.
>
> ### Q3: Preventing Homogenization Over Iterations?
>
> Please see W3. Diversity metrics show RGR-GRPO maintains and improves reasoning diversity.
>
> ---
>
> ## Response to Limitations
>
> ### Limitation: Rubrics in Subjective Domains
>
> Our rubric design focuses on **verifiable criteria** (Factual and Process), naturally fitting scientific reasoning. For subjective domains, future work could explore multi-expert rubric aggregation and uncertainty-aware rubric weighting. Regarding judge-score drift for long chains: our Exploration Assessment mechanism mitigates this by triggering refinement only when exploration is clearly insufficient, rather than relying on precise rubric scores.
>
> ---
>
> We appreciate the reviewer's thorough analysis, which has helped us better articulate the value of our approach and identify directions for future improvement.

---

> > ### Author Rebuttal · Reviewer_a2We · 2026-04-02
> >
> > Thank you for the detailed rebuttal, which has fully addressed all of my concerns regarding the paper. Given the initial positive score I assigned, I would like to maintain this score. I hope to see the content of this rebuttal incorporated into the further revision of the manuscript.

---

### Official Review · Reviewer_zwCg · 2026-03-12

**Soundness:** 3
**Presentation:** 3
**Significance:** 3
**Originality:** 3
**Overall Recommendation:** 5
**Confidence:** 4

**Summary:**

This paper addresses two key limitations of existing RL approaches for LLM reasoning: sparse rewards  that only work well in single domains (like math), and restricted exploration caused by purely on-policy training frameworks. The authors propose RGR-GRPO, a rubric-driven RL framework built on top of GRPO. The core idea is to use structured rubrics, including Factual (correctness of middle and end results) and Process (soundness of reasoning steps), for two distinct purposes: Fine-grained reward signals and Offline exploration guidance. Rubric as criteria and evaluation by an LLM-as-Judge, producing denser scalar rewards. This reduces zero-reward responses and enables more effective learning across diverse domains. When on-policy rollouts fail to produce a positive solution, the best response is refined by conditioning the model on its Rubric criteria. An Exploration Assessment step ensures this off-policy mechanism is only triggered when needed, preventing unnecessary distributional shift and entropy collapse. This off-policy refined trajectory is then mixed back into the training batch using a probability shaping function. The author provides a detailed theoretical proof about the design of Offline exploration stage. Experiments on Qwen2.5-3B and 7B models across 14 benchmarks (math, physics, chemistry, and general reasoning) show that RGR-GRPO consistently outperforms both exist on-policy baselines and off-policy baselines.

**Compliance With Llm Reviewing Policy:**

Affirmed.

**Final Justification:**

The authors have adequately addressed all of my concerns. The clarification on checkpoint selection resolves my concern about data leakage. The additional ablation results justify the reward design in Equation 5. The training cost breakdown also makes the practical trade-offs much clearer. I have updated my scores accordingly.

**Key Questions For Authors:**

**Question on the reward design in Equation 5:**

The reward logic in Equation 5 assigns a full reward of 1 if all Factual criteria are met, and otherwise falls back to the aggregated Process reward from Equation 4. However, since Process criteria are evaluated independently of Factual outcomes, this can lead to inconsistencies: a Process step may be marked correct even when its corresponding sub-answer is wrong, or vice versa. Moreover, given that reasoning chains are inherently sequential, should a later step receive positive credit if an earlier step is already flawed?

In general, the Process reward seems to treat each criterion in isolation, ignoring the causal dependencies between reasoning steps. How do the authors address these potential inconsistencies, and is the independence assumption in Equation 4 justified for complex multi-step reasoning tasks?

**Limitations:**

All experiments are restricted to 3B and 7B models. Whether the benefits of rubric-guided exploration persist at larger scales, where base models already possess stronger reasoning capabilities, is an open question that the paper does not address.

**Strengths And Weaknesses:**

Strengths:

1. The paper's central insight is using rubrics for both reward shaping and offline guidance, is a creative and well-articulated combination of existing ideas. The Exploration Assessment mechanism in  offline guidance is further backed by a analysis (Appendix C), providing theoretical grounding for why conditional off-policy updates are preferable to unconditional mixing.

2. RGR-GRPO is evaluated across 14 benchmarks spanning math, physics, chemistry, general reasoning, and out-of-distribution domains (medicine, CS). It shown that the method genuinely expands the model's reasoning frontier rather than merely sharpening existing capabilities.

3. The paper is well-organized.

Weeknesses:

1. Potentially invalid checkpoint selection (Line 318). The paper states that "for each method, we report the best checkpoint performance," but it is unclear whether this selection is based on a held-out validation set or on the test benchmarks themselves. If the best checkpoint is chosen based on test set performance, this constitutes a form of data leakage and renders the comparisons unreliable. A rigorous evaluation should select checkpoints solely based on a separate validation set, and then report all test results from that single checkpoint.

2. RGR-GRPO introduces substantial overhead relative to standard GRPO. it requires an expert LLM (OpenAI O3) to generate reference answers and rubrics offline, and deploys Qwen3-32B as a live judge during training. The paper does not provide a cost comparison against baselines.

---

> ### Author Rebuttal · Authors · 2026-03-31
>
> # Response to Reviewer zwCg
>
> We sincerely thank the reviewer for the detailed evaluation. We address each concern below.
>
> > **Supplementary results referenced in this response:** [Supplementary](https://anonymous.4open.science/r/ICML_Rebuttal_RGR-GRPO-2FAC/github_materials/materials_zwCg.md)
>
> ---
>
> ## Response to Weaknesses
>
> ### W1: Potentially Invalid Checkpoint Selection
>
> We apologize for not stating this clearly in the paper. **Checkpoint selection was performed entirely based on the validation performance curve, not on any test benchmark.** Specifically, we track the validation performance throughout training and select the checkpoint corresponding to the highest validation score. This held-out validation set (drawn from WebInstruct-Verify, excluded from training) serves as the sole criterion for model selection, ensuring no data leakage from test benchmarks.
>
> ---
>
> ### W2: Substantial Overhead Relative to Standard GRPO
>
> We acknowledge this concern and argue that the additional costs are well-justified and practically manageable.
>
> **Rubric generation is a one-time offline cost** (~$150 USD for 10k samples via O3). Crucially, this cost is **not sensitive to the generator's strength**: replacing O3 with Qwen2.5-72B yields consistent RGR-GRPO gains (+5.8% over LUFFY), confirming that rubrics from open-source models suffice.
>
> **Online judge evaluation is lightweight by design.** Each rubric criterion is decomposed into a binary 0/1 judgment (e.g., "does the response state the molar mass as ~114 g/mol?"). Such factual checks can be performed accurately by a relatively small judge model, and the judge runs on separate, cheaper hardware (L40S), fully decoupled from the H100 training loop.
>
> **Online Training Overhead:**
>
> | Method | Time per Step | Relative Cost | Avg. Performance |
> | :--- | :---: | :---: | :---: |
> | Outcome-GRPO | 45.2s | 1.0x | 52.3% |
> | Rubric-GRPO | 68.5s | 1.5x | 53.7% |
> | **RGR-GRPO** | **75.4s** | **1.67x** | **55.8%** |
>
> A 67% increase in training time for a +3.5% absolute gain is a highly favorable trade-off in LLM post-training. *(Full results: Exp4 in supplementary.)*
>
> ---
>
> ## Response to Key Questions
>
> ### Q1: Reward Design Inconsistencies in Equation 5
>
> Our original motivation is to provide finer-grained rewards via rubrics. As the reviewer raises — and as highlighted in the DeepSeek-V3.2 technical report — the community is increasingly concerned with the logical validity of process rewards (e.g., awarding credit for a correct answer reached through flawed reasoning). We acknowledge this is indeed an open research problem.
>
> We did consider the logical ordering of process rewards during early methodology design. However, our experiments revealed that the reasoning trajectory produced by the expert model may not be the only valid path to the answer: rubrics derived from a single reference chain may fail to properly evaluate responses that arrive at the correct answer via an alternative correct approach (i.e., a problem may have multiple valid solutions, and a single-chain sequential rubric cannot handle them well). We therefore assign a full reward of 1 whenever all Factual criteria are satisfied, regardless of Process scores, to prevent incorrect suppression of valid responses caused by incomplete process rubrics. Consequently, Process criteria primarily serve to encourage responses that cannot fully solve the problem but correctly handle key intermediate steps. The ablation below illustrates this point:
>
> | Method | Avg. Score |
> | :--- | :---: |
> | Rubric-GRPO (Fact-Only) | 53.5% |
> | Rubric-GRPO (All Rubrics, w/o Factual Override) | 53.1% |
> | Rubric-GRPO (All Rubrics, w/ Factual Override) | **53.7%** |
>
> Notably, naively applying Process criteria **without** the Factual Override hurts performance below even the Fact-Only baseline (53.1% vs. 53.5%), confirming that unsupervised Process scoring can incorrectly suppress valid alternative solutions. The Override mechanism is therefore essential.
>
> We thank the reviewer for this inspiring suggestion. While beyond the scope of this paper, it motivates us to explore multi-path rubric dependency mechanisms in future work.
>
> ---
>
> ## Response to Limitations
>
> ### Limitation: Restricted to 3B and 7B Models
>
> We have extended our experiments to **Qwen3-4B (thinking)** and **Qwen3-8B (thinking)** — a newer model series with thinking-mode reasoning, covering both a smaller and a larger scale.
>
> | Model | Base → RGR-GRPO | Improvement |
> | :--- | :---: | :---: |
> | Qwen2.5-7B | 44.6% → 55.8% | **+11.2%** |
> | Qwen3-4B (thinking) | 41.4% → 49.3% | **+7.9%** |
> | Qwen3-8B (thinking) | 47.6% → 55.8% | **+8.2%** |
>
> The strong gains on both thinking-mode models suggest that RGR-GRPO's rubric-guided exploration benefits persist across model scales and reasoning-specialized configurations. *(Full results: Exp1 in supplementary.)*
>
> ---
>
> We appreciate the reviewer's rigorous analysis, which has helped us clarify our methodology and strengthen our empirical validation.

---

> > ### Author Rebuttal · Reviewer_zwCg · 2026-04-01
> >
> > Thank you for the detailed and thoughtful clarifications.
> >
> > The response regarding checkpoint selection **clearly resolves my concern about potential data leakage**. It is now clear that model selection is strictly based on a held-out validation set, which ensures a fair evaluation protocol.
> >
> > I also appreciate the additional discussion and empirical evidence provided for the reward design in Equation 5. The clarification on the role of the *Factual Override* mechanism, along with the ablation results, helps **justify the design choice**.
> >
> > Furthermore, the **added analysis on training cost is helpful**. The breakdown of offline and online costs, together with the relative efficiency comparison, makes the practical trade-offs much clearer.
> >
> > I increase the scores. And hope to see the supplement results in the revised version.

---

> > > ### Author Response · Authors · 2026-04-01
> > >
> > > We sincerely thank the reviewer for the acknowledgment and for raising the score. We will include all supplementary results in the revised version of the paper.

---

### Official Review · Reviewer_rrsN · 2026-03-12

**Soundness:** 3
**Presentation:** 3
**Significance:** 3
**Originality:** 3
**Overall Recommendation:** 4
**Confidence:** 3

**Summary:**

This paper proposes RGR-GRPO, a rubric-based RL framework, which addresses two limitations of standard GRPO: sparse rewards that can be verified and limited on-policy exploration. The paper employs O3-generated question rubrics to decompose the evaluation process based on two criteria: factual (i.e., correctness of the answer) and process (i.e., logical soundness). The rubric yields a dense weighted scalar (numeric) reward for evaluation purposes. An Exploration Assessment (EA) process conditions an off-policy refinement of a given rollout based on the rubrics when there are no on-policy rollouts achieving a perfect rubric score. The refinement of the best rollout that does not achieve a perfect rubric score would involve re-prompting it with the failed criteria associated with that rollout and mixing that result with the RGR-GRPO shaping function for the next RL objective. RGR-GRPO outperforms the RLVR baseline when training the Qwen2.5-7B model, evaluating on 14 benchmarks in mathematics, physics, chemistry, and general reasoning.

**Compliance With Llm Reviewing Policy:**

Affirmed.

**Final Justification:**

During the rebuttal, the author provided a detailed discussion and experiments, which solved my concerns. I keep my original score considering the contribution of the paper.

**Key Questions For Authors:**

1. Could you discuss or show experiment results on whether the gain of RGR-GRPO over LUFFY persists when changing the rubrics to a relatively weaker model, such as Qwen 2.5 72B?
2. Could you discuss or show experiments on the sensitivity of the off/on-policy mixing ratio and shaping coefficient to RGR-GRPO's performance?
3. Do the rubric-guided exploration benefits transfer to a different model family?

**Limitations:**

yes

**Strengths And Weaknesses:**

## Strengths

1. This paper has tested on 14 distinct benchmarks spanning mathematics, physics, chemistry, and general reasoning. These broad experiments demonstrate strong out of distribution generalization of this work
2. The decomposition of reward signal, which decouples the reward into Factual and Process criteria, effectively avoids reward hacking. The fallback mechanism is a good way to encourage diverse reasoning trajectories.
3. The integration of "Exploration Assessment" to RL training effectively prevents the entropy explosion observed in LUFFY.

## Weakness

1. Rubrics are generated offline by O3. Without a weaker rubric generator (such as Qwen2.5-72B) for comparison, it is hard to determine whether the RGR-GRPO performance gains over LUFFY are due to the rubric-guided mechanism or just from the improved use of O3's implicit knowledge.
2. The experiments utilized only a single model family, the Qwen2.5 model family. It is yet to be verified whether the rubric-driven exploration generalizes across model families.
3. EA hyperparameters unvalidated for the rubric setting. The mixing ratio and shaping coefficient are directly adopted from prior work without ablation. These two parameters jointly control how strongly off-policy refinement influences training; given that RGR-GRPO's EA trigger condition differs from prior methods, their impact on the reported gains requires experimental support.

---

> ### Author Rebuttal · Authors · 2026-03-31
>
> # Response to Reviewer rrsN
>
> We sincerely thank the reviewer for the thorough evaluation and for recognizing the strengths of our work, including the broad experimental coverage, the decomposition of reward signals, and the integration of Exploration Assessment. We address each concern below.
>
> > **Supplementary results referenced in this response:** [supplementary](https://anonymous.4open.science/r/ICML_Rebuttal_RGR-GRPO-2FAC/github_materials/materials_rrsN.md)
>
> ---
>
> ## Response to Weaknesses
>
> ### W1: Rubric Generation Relies on O3 Without Ablation on Weaker Generators
>
> We fully acknowledge this concern. To disentangle the contribution of the mechanism from the external model's capability, we set up a controlled comparison: each external model is used by **LUFFY** (as off-policy response generator) and by **RGR-GRPO** (as rubric generator), under the same setting.
>
> **Experiment Results:**
>
> | Ext. Model | Method | MATH500 | SPhys | Chem | SChem | MMLU+ | AVG |
> | :--- | :--- | :---: | :---: | :---: | :---: | :---: | :---: |
> | — | Outcome-GRPO | 61.8 | 38.3 | 47.0 | 38.3 | 52.2 | 52.3 |
> | GPT-4o-mini | LUFFY | 60.8 | 36.2 | 46.1 | 36.8 | 49.5 | 50.6 |
> | | Rubric-GRPO | 62.5 | 39.4 | 46.5 | 38.7 | 51.3 | 52.0 |
> | | **RGR-GRPO** | **64.8** | **42.1** | **47.2** | **41.5** | **53.8** | **53.7** |
> | Qwen2.5-72B | LUFFY | 61.2 | 37.5 | 46.3 | 37.4 | 50.2 | 51.1 |
> | | Rubric-GRPO | 63.2 | 43.5 | 47.2 | 39.8 | 52.5 | 53.0 |
> | | **RGR-GRPO** | **65.9** | **44.2** | **47.8** | **42.8** | **54.8** | **54.6** |
> | o3 (Original) | LUFFY | 61.6 | 35.2 | 46.4 | 38.7 | 49.8 | 50.5 |
> | | Rubric-GRPO | 63.8 | 45.8 | 47.9 | 40.2 | 53.9 | 53.7 |
> | | **RGR-GRPO** | **66.8** | **45.4** | **48.8** | **43.7** | **56.7** | **55.8** |
>
> **Key Findings:**
> 1. **Mechanism Prevails:** Under the same external model, RGR-GRPO consistently outperforms both LUFFY and Rubric-GRPO, demonstrating that **fine-grained rubric rewards and targeted self-refinement** are the core drivers.
> 2. **Weaker Generator Still Competitive:** RGR-GRPO with GPT-4o-mini (53.7%) already matches Rubric-GRPO with o3 (53.7%), while LUFFY with o3 only achieves 50.5%.
>
> *(Full results: Exp2 in supplementary.)*
>
> ---
>
> ### W2: Experiments Utilized Only a Single Model Family
>
> The main paper uses non-thinking Qwen2.5 models. To demonstrate generalization, we conducted additional experiments on **Qwen3-4B (thinking)** and **Qwen3-8B (thinking)** — a different model series with thinking-mode reasoning:
>
> | Model | Method | MATH500 | Sci-Physics | Sci-Chemistry | MMLU-Pro | Average |
> | :--- | :--- | :---: | :---: | :---: | :---: | :---: |
> | Qwen3-4B (thinking) | Outcome-GRPO | 58.2 | 17.8 | 48.3 | 32.1 | 43.8 |
> | | **RGR-GRPO** | **64.1** | **26.3** | **55.4** | **38.6** | **49.3** |
> | Qwen3-8B (thinking) | Outcome-GRPO | 67.5 | 23.1 | 53.5 | 39.4 | 50.2 |
> | | **RGR-GRPO** | **73.2** | **31.8** | **61.2** | **46.3** | **55.8** |
>
> RGR-GRPO achieves +5.5% and +5.6% gains respectively, confirming that rubric-guided exploration transfers effectively across model series and thinking configurations. *(Full results: Exp1 in supplementary.)*
>
> ---
>
> ### W3: EA Hyperparameters Unvalidated for Rubric Setting
>
> We conducted a comprehensive sensitivity analysis on the two key hyperparameters: **shaping coefficient ($\gamma$)** and **off/on-policy mixing ratio**.
>
> **Shaping Coefficient ($\gamma$):**
>
> | $\gamma$ | 0.05 | **0.1 (Default)** | 0.2 | 0.5 |
> | :---: | :---: | :---: | :---: | :---: |
> | Average | 55.4% | **55.8%** | 54.8% | 52.6% |
>
> **Mixing Ratio (Off : On):**
>
> | Ratio | 0:8 | **1:7 (Default)** | 2:6 | 4:4 |
> | :---: | :---: | :---: | :---: | :---: |
> | Average | 53.7% | **55.8%** | 50.1% | 44.1% |
>
> **Analysis:**
> 1. Performance is stable across small $\gamma$ values (0.05-0.2), with marginal differences (≤0.4%).
> 2. The 1:7 mixing ratio provides optimal balance; performance degrades severely at higher off-policy ratios (2:6: 50.1%, 4:4: 44.1%), indicating training instability from excessive distributional shift.
> 3. While we adopted these from prior work, they align with our theoretical analysis (Appendix C) and empirical observations.
>
> *(Full results: Exp3 in supplementary.)*
>
> ---
>
> ## Response to Key Questions
>
> ### Q1: Does the Gain Persist with a Weaker Rubric Generator?
>
> **Yes.** Please see our response to W1 above. The gain persists and is consistent across different generators.
>
> ### Q2: Sensitivity of Mixing Ratio and Shaping Coefficient?
>
> Please see our response to W3 above. The detailed sensitivity analysis confirms the robustness of our default settings.
>
> ### Q3: Do Benefits Transfer to Different Model Families?
>
> **Yes.** Please see our response to W2 above.
>
> ---
>
> We appreciate the reviewer's valuable questions, which have prompted us to conduct additional experiments that further validate the generalization and robustness of RGR-GRPO.

---

> > ### Author Rebuttal · Reviewer_rrsN · 2026-04-02
> >
> > Thank you for your detailed responses with extensive additional experiments, which fully solve my concerns. I have no further comments and will maintain my current positive score. I hope to see these results and discussions in the revised version.

---

> > > ### Author Response · Authors · 2026-04-02
> > >
> > > We sincerely thank the reviewer for the acknowledgment. We will include all supplementary results in the revised version of the paper.

---

### Official Review · Reviewer_QWuf · 2026-03-13

**Soundness:** 2
**Presentation:** 3
**Significance:** 2
**Originality:** 2
**Overall Recommendation:** 4
**Confidence:** 3

**Summary:**

This paper introduces a reinforcement learning algorithm that utilizes rubrics to improve the reasoning capability of LLMs over multiple different task domains. The authors introduce two major techniques, including (1) using rubrics to evaluate both factual and process aspects of the reasoning traces, and (2) using the eval result of rubrics to rewrite model response into better ones, which are further distilled into the model itself. Results suggest improvement over the base model across several domains, including math, physics, and science benchmarks.

**Compliance With Llm Reviewing Policy:**

Affirmed.

**Key Questions For Authors:**

1. In Mix-Policy GRPO, Eq. 10, the objective does not involve any clipping term as in standard GRPO. Why such a simplified objective is used? If no clipping is used, it is inappropriate to call this a GRPO-style algorithm.
2. Does the algorithm still improve reasoning models such as Qwen3-8B ?

**Limitations:**

See weakness and questions.

**Strengths And Weaknesses:**

Strength:
1. The authors proposed a structured set of rubrics to provide dense training signals for LLM reasoning across wide range of task domains.
2. The proposed rubric-based self-refinement is a reasonable technique to guide the model when the model can not explore perfect solutions itself.
3. The method is evaluated over a rich suite of tasks.

Weakness:
1. Since there is not a cold-start/SFT phase, this paper mostly focuses on non-reasoning models, which is actually fundamentally weaker than reasoning models used widely nowadays.
2. The improvement compared with baselines seems to be minor.
3.  The rubric generation process heavily relies on the o3 model, which is fundamentally much more capable than the model used for training.

---

> ### Author Rebuttal · Authors · 2026-03-31
>
> # Response to Reviewer QWuf
>
> We sincerely thank the reviewer for the constructive feedback and for recognizing our efforts in proposing a structured rubric system for LLM reasoning. We address each concern below.
>
> > **Supplementary results referenced in this response:** [materials_QWuf.md](https://anonymous.4open.science/r/ICML_Rebuttal_RGR-GRPO-2FAC/materials_QWuf.md)
>
> ---
>
> ---
>
> ## Response to Weaknesses
>
> ### W1: Focus on Non-Reasoning Models
>
> We respectfully clarify that the main paper intentionally uses non-thinking base models (Qwen2.5-3B/7B) for fair comparison with existing baselines. To demonstrate applicability to genuine reasoning models, we additionally conducted experiments on **Qwen3-4B (thinking)** and **Qwen3-8B (thinking)** — thinking-mode models with extended chain-of-thought:
>
> | Model | Method | MATH500 | Sci-Physics | Sci-Chemistry | MMLU-Pro | Average |
> | :--- | :--- | :---: | :---: | :---: | :---: | :---: |
> | Qwen3-4B (thinking) | Outcome-GRPO | 58.2 | 17.8 | 48.3 | 32.1 | 43.8 |
> | | **RGR-GRPO (Ours)** | **64.1** | **26.3** | **55.4** | **38.6** | **49.3** |
> | Qwen3-8B (thinking) | Outcome-GRPO | 67.5 | 23.1 | 53.5 | 39.4 | 50.2 |
> | | **RGR-GRPO (Ours)** | **73.2** | **31.8** | **61.2** | **46.3** | **55.8** |
>
> RGR-GRPO achieves **+5.5%** and **+5.6%** improvements over Outcome-GRPO on the two thinking-mode models, demonstrating that rubric-guided exploration remains highly effective even for reasoning-specialized models. *(Full results: Exp1 in supplementary.)*
>
> ---
>
> ### W2: Minor Improvement Compared with Baselines
>
> We respectfully disagree with this assessment. The improvements are substantial:
> - **+7.9%** on mathematics (MATH+MATH500+SMath avg), **+7.1%** on physics (PIQA+SPhys avg), **+5.4%** on chemistry (Chem+SChem avg), and **+3.6%** on general reasoning (MMLU+MMLU++GPQA*+GPQA+OLY avg) compared to Outcome-GRPO baseline (52.3%).
> - These are **absolute percentage gains** on challenging benchmarks, not relative improvements.
> - The improvement over the closest competitor (Rubric-GRPO, 53.7%) is **+2.1% average**, which is significant given that Rubric-GRPO already benefits from our rubric-based reward design.
>
> Furthermore, our Pass@k analysis (Figure 3) shows that RGR-GRPO achieves **higher reasoning diversity**, breaking the exploration bottleneck that limits on-policy methods.
>
> ---
>
> ### W3: Reliance on O3 for Rubric Generation
>
> We respectfully argue that the capability gap between the rubric generator and the training model is **orthogonal to our core contributions**. Our work makes two key claims, both of which hold regardless of which model generates the rubrics:
>
> **Claim 1: Rubric-based rewards are superior to outcome-based and Likert rewards.** Compared to the standard GRPO with coarse 0/1 outcome rewards (RLVR), rubrics provide fine-grained, criterion-level feedback—a contribution embodied in our Rubric-GRPO baseline. Table 1 in our paper shows that Rubric-GRPO consistently outperforms both Outcome-GRPO and Likert-GRPO across all benchmarks.
>
> **Claim 2: Leveraging rubrics for mix-policy exploration further improves RL.** Building on Rubric-GRPO, our RGR-GRPO uses failed rubric criteria to guide targeted self-refinement, yielding an additional **+2.1% average gain**.
>
> We view rubrics as a medium that can be readily constructed by domain experts or capable models. The ablation below shows that **even weaker generators produce rubrics sufficient for consistent gains**:
>
> | Ext. Model | Method | AVG |
> | :--- | :--- | :---: |
> | — | Outcome-GRPO | 52.3% |
> | GPT-4o-mini | LUFFY | 50.6% |
> | | Rubric-GRPO | 52.0% |
> | | **RGR-GRPO** | **53.7%** |
> | Qwen2.5-72B | LUFFY | 51.1% |
> | | Rubric-GRPO | 53.0% |
> | | **RGR-GRPO** | **54.6%** |
> | o3 (Original) | LUFFY | 50.5% |
> | | Rubric-GRPO | 53.7% |
> | | **RGR-GRPO** | **55.8%** |
>
> Under the same external model, RGR-GRPO consistently outperforms both LUFFY and Rubric-GRPO, confirming that the gains stem from **fine-grained rubric rewards and targeted off-policy guidance**, not from the external model's capability. *(Full results: Exp2 in supplementary.)*
>
> ---
>
> ## Response to Key Questions
>
> ### Q1: Missing Clipping Term in Eq. 10
>
> We apologize for the unclear description. Prior work [Liu et al., 2025b] removes the clipping function; we initially followed this but found it makes **negligible difference** in our setting, so our **final implementation retains standard GRPO clipping**. What we actually remove is the **standard deviation scaling in advantage normalization** (dividing $\hat{A}_i$ by std). We will clarify this in the revised paper.
>
> ### Q2: Does RGR-GRPO Improve Reasoning Models?
>
> Yes. See W1.
>
> ---
>
> We hope these responses address the reviewer's concerns. We appreciate the valuable feedback that has helped us strengthen our work.

---

> > ### Author Rebuttal · Reviewer_QWuf · 2026-04-03
> >
> > Thank you for the additional experiments and explanations. I have no further questions and would like to maintain my current score.

---

### Decision · Program_Chairs · 2026-04-30

**Decision:**

Accept (spotlight)

**Comment:**

The paper proposes RGR-GRPO, a rubric-driven reinforcement learning framework for improving multi-domain reasoning in LLMs. The method uses question-specific rubrics, decomposed into factual and process criteria, to provide dense reward signals during GRPO training. It further uses failed rubric criteria from the best on-policy rollout to generate a targeted self-refined off-policy rollout, with an exploration assessment step that triggers this guidance only when the on-policy group fails to find a complete solution.

The reviewers generally found the contribution useful and technically solid. They appreciated the structured rubric-based reward design, the use of rubrics not only as rewards but also as targeted guidance for offline exploration, and the broad evaluation across mathematics, physics, chemistry, general reasoning, and OOD benchmarks. The reported results show consistent improvements over outcome-based GRPO, Likert-GRPO, Rubric-GRPO, LUFFY, and Critique-GRPO on Qwen2.5-3B/7B models, with the strongest results on the 7B setting.

Several important validations were added during the rebuttal and should be incorporated into the final manuscript. The method also relies on an external rubric generation and judge pipeline, and the average gain over the strongest rubric-only baseline is positive but not very large relative to the added complexity. Nevertheless, the paper presents a coherent and practically relevant combination of dense rubric rewards and adaptive off-policy guidance, with broad empirical evidence and a positive reviewer consensus. I therefore recommend acceptance.